# Mirror, Mirror of the Flow: How Does Regularization Shape Implicit Bias?

Tom Jacobs [1]   Chao Zhou [1]   Rebekka Burkholz [1]

## Abstract

Implicit bias plays an important role in explaining how overparameterized models generalize well. Explicit regularization like weight decay is often employed in addition to prevent overfitting. While both concepts have been studied separately, in practice, they often act in tandem. Understanding their interplay is key to controlling the shape and strength of implicit bias, as it can be modified by explicit regularization. To this end, we incorporate explicit regularization into the mirror flow framework and analyze its lasting effects on the geometry of the training dynamics, covering three distinct effects: positional bias, type of bias, and range shrinking. Our analytical approach encompasses a broad class of problems, including sparse coding, matrix sensing, single-layer attention, and LoRA, for which we demonstrate the utility of our insights. To exploit the lasting effect of regularization and highlight the potential benefit of dynamic weight decay schedules, we propose to switch off weight decay during training, which can improve generalization, as we demonstrate in experiments.

## 1. Introduction

Regularization is a fundamental technique in machine learning that helps control model complexity, prevent overfitting and improve generalization (Kukačka et al., 2017). We focus on the interplay between two major categories of regularization: explicit regularization and implicit bias. We introduce both concepts within a general minimization problem. Consider the objective function $f\colon \mathbb{R}^n \to \mathbb{R}$ to be minimized with respect to $x$:

$$\min_{x \in \mathbb{R}^n} f(x). \tag{1}$$

In the context of explicit regularization, a penalty term $h(x)$ is incorporated into the objective function, directly prevent-

---

[1]CISPA Helmholtz Center for Information Security, Saarbrücken, Germany. Correspondence to: Tom Jacobs <tom.jacobs@cispa.de>.

*Proceedings of the 42nd International Conference on Machine Learning*, Vancouver, Canada. PMLR 267, 2025. Copyright 2025 by the author(s).

ing the learning algorithm from overfitting (Goodfellow et al., 2016), as follows:

$$\min_{x \in \mathbb{R}^n} f(x) + \alpha h(x), \tag{2}$$

where $\alpha$ controls the trade-off between the objective and the penalty. This approach regulates the model capacity (Dai et al., 2021) and encourages simpler solutions that are more likely to generalize well to unseen data (Tian & Zhang, 2022). Common explicit regularization methods include $L_1$ (LASSO) and $L_2$ (weight decay) (Bishop & Nasrabadi, 2006). The effectiveness of explicit regularization techniques has been demonstrated across various machine learning paradigms (Arpit et al., 2016), including supervised learning, unsupervised learning and reinforcement learning.

Implicit bias (Gunasekar et al., 2017; Woodworth et al., 2020; Li et al., 2022; Sheen et al., 2024; Vasudeva et al., 2024; Tarzanagh et al., 2023; Jacobs & Burkholz, 2025) can be considered as an inherent aspect of the model design and optimizer that does not require explicit modifications of the objective function. The goal of characterizing the implicit bias is to understand how overparameterization impacts the training dynamics and, consequently, model selection. For example, in the presence of many global minima, optimization algorithms like gradient descent inherently converge towards low-norm solutions (Woodworth et al., 2020; Pesme et al., 2021), which impacts model properties such as generalization (Belkin et al., 2019) and memorization (Radhakrishnan et al., 2020).

Implicit bias is often associated with a mirror flow (Karimi et al., 2024; Li et al., 2022), which results from a reparameterization of $f$ by setting $x = g(w)$, where $w \in M$ and $M$ is a smooth manifold. It is important to highlight a fundamental distinction between the explicit regularization in the original space and the mirror flow with the objective function, formulated as follows:

$$\min_{w \in M} f(g(w)) + \alpha h(w). \tag{3}$$

The explicit regularizer $h$ now acts on the parameters $w$ instead of $x = g(w)$. Our main goal is to understand how the explicit regularization $h(w)$ affects the implicit bias, thereby shaping the effective regularization in the original parameter space $x$. To achieve this, we analyze their interplay by integrating explicit regularization into the mirror flow framework.

Typically, the nature and strength of implicit bias remain constant throughout training as they are inherently determined by the model parameterization. For instance, it has been shown that specific forms of overparametrization lead to low-rank or sparse solutions (Arora et al., 2019; Pesme et al., 2021; Sheen et al., 2024; Woodworth et al., 2020; Gunasekar et al., 2017), revealing a bias towards sparsity in particular settings. Nevertheless, factors such as small initialization, large learning rates and noise are needed to obtain this sparsity bias, without guarantees. However, the inherent bias can degrade performance if it does not fit to the learning task. Our key insight to overcome this issue is that implicit bias can be adapted and controlled by explicit and potentially dynamic regularization, which induces a time-dependent mirror flow. To analyze the resulting optimization problem within the extended mirror flow framework (Li et al., 2022) and obtain convergence and optimality results, we provide sufficient conditions for the reparameterization $g$ and explicit regularization $h$. Additionally, we characterize the regularization $h$ in terms of $g$ to understand their interplay and impact on the Legendre function, which is associated with the implicit bias.

Concretely, we identify three distinct effects:

- Type of bias: the explicit regularization changes the shape of the Legendre function and thus the implicit bias. For example, the shape changes from an $L_2$ norm to $L_1$ norm.

- Positional bias: in the standard case without explicit regularization, the global minimum of the Legendre function corresponds to the parameter initialization (Li et al., 2022). Explicit regularization shifts this global minimum, gradually moving it closer to zero during training.

- Range shrinking: the explicit regularization shrinks the range of the attainable values for the Legendre function. For example, the $L_1$ norm of the parameters becomes stationary during training.

The three effects are illustrated in Figure. 1. They all have a lasting impact on implicit bias, as they change the geometry of the training dynamics.

Weight decay provides an illustrative example of an explicit regularization that has a desirable impact on the training dynamics (D'Angelo et al., 2023). While (Dai et al., 2021) studied the effect of constant regularization on model capacity, and (Khodak et al., 2021; Kobayashi et al., 2024) empirically observed that weight decay leads to sparsity bias for quadratic reparameterizations, our focus lies on understanding the effects of both constant and dynamic explicit regularization on the training dynamics and implicit bias.

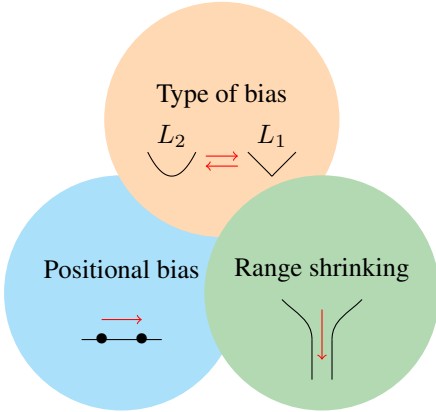

*Figure 1.* Illustration of three established effects of explicit regularization ($\rightarrow$) on implicit bias.

This offers a deeper theoretical insight into the interplay between implicit bias and explicit regularization.

Our theoretical framework has multiple application-relevant implications. As examples, we explore sparse coding, matrix sensing, attention mechanisms in transformers, and Low-Rank Adaptation (LoRA) through experiments. By switching off weight decay during training, we demonstrate the positive impact of dynamic regularization on generalization performance and analyze its effect on implicit bias.

**Contributions:**

- We establish sufficient conditions for incorporating different types of explicit regularization into the mirror flow framework and characterize their effects, focusing on three key impacts on implicit bias: positional bias shift, type of bias, and range shrinking.

- We propose a systematic procedure for identifying appropriate regularizations and establish general convergence and optimality results. These results provide guidance on how to manage the above effects by adjusting the explicit regularization.

- We gain the insight that explicit regularization controls the strength of implicit sparsification and has a lasting effect by changing the geometry of the training dynamics.

- We highlight the positive impact of dynamic regularization and the resulting implicit bias through experiments such as sparse coding, matrix sensing, attention mechanism, and LoRA in fine-tuning large language models.

## 2. Related Work

**Regularization**  There are multiple ways to regularize in machine learning. Some of the most widely used techniques include weight decay (D'Angelo et al., 2023; Krogh

& Hertz, 1991), data augmentation (Cubuk et al., 2020; Orvieto et al., 2023), dropout (Srivastava et al., 2014), and batch normalization (Ioffe & Szegedy, 2015). Weight decay, or $L_2$ regularization, discourages large weights to mitigate overfitting and induces a desirable change in training dynamics. This change can be effectively captured using the time-dependent mirror flow framework that we extend. As another example, dynamic weight decay has been proposed for ADAM to keep the gradient norms in check (Xie et al., 2023). In comparison, we analyze the effect of more general dynamic regularization on the implicit bias of gradient flow.

**Implicit Bias** The implicit bias is a well-studied phenomenon (Woodworth et al., 2020; Gunasekar et al., 2017; 2018; Li et al., 2022), which has primarily been characterized within the mirror flow framework, a well-established concept in convex optimization (Alvarez et al., 2004; Beck & Teboulle, 2003; Rockafellar & Fenchel, 1970; Boyd & Vandenberghe, 2009), which we extend by introducing explicit regularization that can induce a time-dependent Legendre function. Moreover, for convergence guarantees the time-dependent Legendre function needs to satisfy additional assumptions, i.e., it needs to be a time-dependent Bregman function. For this class of functions we show convergence with decaying regularization. A mirror flow can be interpreted as a gradient flow on a Riemannian manifold (Li et al., 2022; Alvarez et al., 2004), which has also been studied in stochastic gradient descent (SGD) (Pesme et al., 2021; Even et al., 2023; Lyu & Zhu, 2023) context. The main observation is that large learning rates and stochastic noise from SGD have a generalization benefit by inducing sparsity, although uncontrollable. We derive a similar but controllable benefit of explicit regularization. Still, it is possible to also combine stochastic noise and a large learning rate with our framework. Discrete versions of mirror flow (Sun et al., 2022) have led to novel algorithmic designs (Raj & Bach, 2021; Gonzalez et al., 2024; Azizan et al., 2022). Time-dependent mirror descent, in comparison, is underexplored, except for an analysis of its intrinsic properties and an application to continuous sparsification (Radhakrishnan et al., 2021; Jacobs & Burkholz, 2025). Our framework covers multiple application relevant architectures and more general cases.

**Applications of the Mirror Flow Framework** The mirror flow framework has been applied to various architectures, including attention mechanisms in transformers (Vaswani, 2017; Vasudeva et al., 2024; Sheen et al., 2024; Tarzanagh et al., 2023; Julistiono et al., 2024; Pesme et al., 2024), matrix factorization (Li et al., 2021; Gunasekar et al., 2017; 2018) and diagonal linear networks (Li et al., 2022; Pesme et al., 2021; Woodworth et al., 2020). The implicit bias of deep matrix factorization has also been analyzed with gradient flow methods (Marion & Chizat, 2024; Arora et al.,

2019). Accordingly, the flow tends to be implicitly biased towards solutions with low rank or nuclear norm. We show that dynamic explicit $L_2$ regularization can further enhance this effect in the context of quadratic overparameterization. This is illustrated through experiments on transformer networks. Moreover, we identify the inherent bias of LoRA (Hu et al., 2022; Wan et al., 2024) and delve into the associated challenges. This is especially of interest, as LoRA has gained significant popularity in the field of large language models (LLMs), as it allows for cost-effective finetuning. Another application is sparse coding (SC) which is similar to diagnonal linear networks with explicit regularization. This representation technique is widely employed in signal processing and pattern recognition (Zhang et al., 2015). The core principle of SC is to find a sparse representation by imposing constraints, typically using the $L_0$-norm. However, this formulation leads to an NP-hard problem (Tropp, 2004). An alternative strategy relaxes the constraint, transforming the original problem into a convex, albeit non-smooth optimization task. Proximal algorithms have proven effective to solve these non-smooth problems (Daubechies et al., 2004). Similarly, reparameterization with explicit regularization can be used to solve this.

## 3. Theory: Integrating Explicit Regularization in the Extended Mirror Flow Framework

We begin by reviewing the theoretical background on reparameterizations and when they induce mirror flows. Our main result, Theorem 3.6, integrates explicit regularization into the mirror flow framework. Building on this, we explore key implications, including a geometric interpretation of the interaction between implicit bias and explicit regularization. Using this interpretation and assuming the Legendre function $R$ is a Bregman function, we extend convergence results to the time-dependent setting by introducing the contracting property (Definition 3.8, Theorem 3.10). We also prove optimality in underdetermined linear regression (Theorem 3.11). To apply our theory in practice, we show how to choose an explicit regularizer $h$ for a given reparameterization $g$, often determined by neural network design. We characterize $h$ for known reparameterizations (Woodworth et al., 2020; Pesme et al., 2021; Gunasekar et al., 2017) and examine their practical effects—type of bias, positional bias, and range shrinking—in Section 4.

### 3.1. Preliminaries

To analyze the impact of regularization on the training dynamics of deep neural networks, we start from the gradient flow for our general optimization problem in Eq. (1). We assume $f \in C^1(\mathbb{R}^n, \mathbb{R})$ to be a continuously differentiable objective function. The corresponding gradient flow is:

$$dw_t = -\nabla_w f(g(w_t))dt, \qquad w_0 = w_{init},$$

where $\nabla_w$ is the gradient with respect to $w$ and $g \in C^1(M, \mathbb{R}^n)$. For a specific choice of $g$, reparameterizing the loss function $f$ leads to a mirror flow with a related implicit bias. For this we recall two definitions that characterize the parameterization. Moreover, we define the Legendre function needed to define the mirror flow.

**Definition 3.1.** (Regular Parameterization, Definition 3.4 (Li et al., 2022)) Let $M$ be a smooth submanifold of $\mathbb{R}^D$. A regular parameterization $g : M \to \mathbb{R}^n$ is a $C^1$ parameterization such that the Jacobian $\partial G(w)$ is of rank $n$ for all $w \in M$.

This ensures that the gradient flow for $x_t = g(w_t)$ does not have an additional null space i.e. the gradient flow can not get stuck due to reparameterization. For the second definition, we first need to define the Lie bracket operator where $\partial$ is the Jacobian operator.

**Definition 3.2.** (Lie Bracket, Definition 3.4 (Li et al., 2022)) Let $M$ be a smooth submanifold of $\mathbb{R}^D$. Given two $C^1$ vector fields $X, Y$ on $M$, we define the Lie Bracket of $X$ and $Y$ as $[X, Y](w) := \partial Y(w)X(w) - \partial X(w)Y(w)$.

**Definition 3.3.** (Commuting Parameterization, Definition 4.1 (Li et al., 2022)) Let $M$ be a smooth submanifold of $\mathbb{R}^D$. A $C^2$ parameterization $g : M \to \mathbb{R}^d$ is commuting in a subset $S \subset M$ iff for any $i, j \in [n]$, the Lie bracket $[\nabla g_i, \nabla g_j](w) = 0$ for all $w \in S$. Moreover, we call $g$ a commuting parameterization if it is commuting in $M$.

Definition 3.3 ensures appropriate eigen basis alignment. We now introduce the Legendre function which governs the mirror flow dynamics.

**Definition 3.4.** (Legendre Function, Definition 3.8 ((Li et al., 2022))) Let $R : \mathbb{R}^d \to \mathbb{R} \cup \{\infty\}$ be a differentiable convex function. We say $R$ is a Legendre function when the following holds:

- $R$ is strictly convex on int(dom$R$).

- For any sequence $\{x_i\}_{i=1}^{\infty}$ going to the boundary of dom$R$, $\lim_{i \to \infty} \|\nabla R(x_i)\|_{L_2}^2 = \infty$.

Appendix A summarizes the main aspects of the mirror flow framework (Li et al., 2022), which explains their relationship. Formally, let the reparameterization $g$ be regular (Definition 3.1), commuting (Definition 3.3) and satisfy Assumption A.6. Then, by Theorem A.7, there is an Legendre function $R : \mathbb{R}^n \to \mathbb{R}$ (Definition 3.4) that follows the dynamics:

$$d\nabla_x R(x_t) = -\nabla_x f(x_t)dt, \qquad x_0 = g(w_{init}). \quad (4)$$

The Legendre function is associated with the implicit bias of the optimization. For example, $R$ can be the hyperbolic entropy studied in (Pesme et al., 2021; Woodworth et al., 2020;

Wu & Rebeschini, 2021). Depending on the initialization of the reparameterization, the hyperbolic entropy is equivalent to either $L_2$ or $L_1$ implicit regularization. A Legendre function $R$ that resembles an $L_1$ regularization is associated with the so-called feature learning regime, which has been argued to improve generalization performance. Accordingly, it presents a positive impact of overparameterization on deep learning.

Notably, in the presence of explicit regularization, the Legendre function $R$ can change over time, which has been recognized by Jacobs & Burkholz (2025) with the specific goal to exploit the implicit bias for gradual sparsification. (Lyu et al., 2024) has analyzed how small constant weight decay impacts implicit bias to study its effect on Grokking, but has done so outside of the mirror flow framework. We allow for dynamic and possibly large regularization of relatively general form. While the implicit bias can change dynamically also for different reasons like large learning rate and stochastic noise as in (Pesme et al., 2021; Lyu & Zhu, 2023), we focus on dynamic explicit regularization to control this change.

### 3.2. Main Result

We characterize the interplay between explicit regularization and implicit bias by a time-dependent Legendre function. In the setting of Eq. (3) with reparameterization $g \in C^1(M, \mathbb{R}^n)$ and explicit regularization $h \in C^1(M, \mathbb{R})$, we allow the regularization strength $\alpha$ to vary over time during the gradient flow, as indicated by an index $\alpha_t$. This induces the following gradient flow:

$$dw_t = -\left(\nabla_w f(g(w_t)) + \alpha_t \nabla_w h(w_t)\right) dt, \quad w_0 = w_{init}.$$

To rigorously define the corresponding time-dependent mirror flow, we define a parameterized Legendre function based on Definition 3.4.

**Definition 3.5.** Let $A$ be a subset of $\mathbb{R}$. A parameterized Legendre function is $R_a : \mathbb{R}^n \to \mathbb{R}^n$ such that for all $a \in A$, $R_a$ is a Legendre function (Definition 3.4).

The next theorem is our main result and builds on Definition 3.5 and Theorem A.7 in the appendix.

**Theorem 3.6.** *Let $(g, h): M \to \mathbb{R}^{n+1}$ be regular and commuting reparameterization satisfying Assumption A.6. Then there exists a time-dependent Legendre function $R_a$ such that*

$$d\nabla_x R_{a_t}(x_t) = -\nabla_x f(x_t)dt, \qquad x_0 = g(w_{init}), \quad (5)$$

*where $a_t = -\int_0^t \alpha_s ds$. Moreover, $R_{a_t}$ only depends on the initialization $w_{init}$ and the reparameterization $g$ and regularization $h$, and is independent of the loss function $f$.*

Proof. See Theorem B.1 in the appendix. The main steps of the proof are: 1) We apply Theorem 4.9 (Li et al., 2022)

to the time-dependent loss function $L_t(x, y) = f(x) + \alpha_t y$ with the reparameterization $x = g(w)$ and explicit regularization $y = h(w)$ to get the resulting mirror flow with Legendre function $R(x, y)$. 2) $R$ is strictly convex. We utilize this to show that $y \to \partial_y R(x, y)$ is invertible. 3) We use the fact that the mirror flow for $y_t$ is defined by $\partial_y R(x_t, y_t) = a_t$, where $a_t = -\int_0^t \alpha_s ds$. Next, we plug in the inverse $y_t = Q(x_t, a_t)$ into $\nabla_x R(x_t, y_t)$ to get an expression for the gradient of the time-dependent Legendre function $R$. This leads to an equation for the time-dependent mirror flow $\nabla_x R(x_t, Q(x_t, a_t)) = \mu_t$, where $\mu_t = -\int_0^t \nabla_x f(x_s) ds$. 4) In the final step, we show that $\nabla_x R(x, Q(x, a))$, where $\nabla_x$ is the derivative with respect to the first entry, is the gradient of a Legendre function for $a$ fixed.

Theorem 3.6 characterizes the training dynamics of reparameterization with regularization. This leads to an additional geometric interpretation, which we use next.

### 3.3. Geometric Interpretation

A mirror flow can be interpreted as a gradient flow on a Riemannian manifold (Li et al., 2022; Alvarez et al., 2004). If a Legendre function $R$ induces a mirror flow, the iterates $x_t$ follow the dynamics:

$$dx_t = -\left(\nabla_x^2 R(x_t)\right)^{-1} \nabla_x f(x_t) dt \qquad x_0 = g(w_{init}), \tag{6}$$

where the manifold metric is given by $\left(\nabla_x^2 R\right)^{-1}$. Accordingly, Theorem 3.6 leads to a new geometric interpretation of a regularization. Specifically, $x_t$ evolves as follows:

$$dx_t = -\left(\nabla_x^2 R_{a_t}(x_t)\right)^{-1} \left(\nabla_x f(x_t) + \alpha_t \nabla_x y_t\right) dt, \tag{7}$$

with initialization $x_0 = g(w_{init})$ and $y_0 = h(w_{init})$, where $y_t$ is defined as in Theorem 3.6. Thus, the effect of regularization on the training dynamics is described by a changing Riemannian metric, where the metric evolves according to the time-dependent Legendre function $R_{a_t}$.

In practice, we can steer $a_t$ and thus influence the manifold. Another perspective on this is that the effect of explicit regularization is stored in the time-dependent Legendre function $R_{a_t}$. Therefore, explicit regularization has a lasting effect on the training dynamics, even after it has been turned off, for instance. This creates a novel connection between explicit regularization and implicit bias. Also past regularization influences future implicit bias by shaping the geometry.

**Convergence** The geometric interpretation not only provides valuable intuition but also helps us to show convergence of the mirror flow for time-dependent Bregman functions $R_{a_t}$. A Bregman function is defined as follows:

**Definition 3.7.** (Bregman function, Definition 4.1 (Alvarez

et al., 2004)) A function $R$ is called a Bregman function if it satisfies the following properties:

- $\text{dom} R$ is closed. $R$ is strictly convex and continuous on $\text{dom} R$. $R$ is $C^1$ on $\text{int}(\text{dom} R)$).

- For any $x \in \text{dom} R$ and $\gamma \in \mathbb{R}$, $\{y \in \text{dom} R | D_R(x, y) \leq \gamma\}$ is bounded.

- For any $x \in \text{dom} R$ and sequence $\{x_i\}_{i=1}^\infty \subset \text{int}(\text{dom} R)$ such that $\lim_{i \to \infty} x_i = x$, it holds that $\lim_{i \to \infty} D_R(x, x_i) \to 0$.

Using Definition 3.7, we can define the parameterized Bregman function next:

**Definition 3.8.** Let $A$ be a subset of $\mathbb{R}$. A parameterized Bregman function is $R_a : \mathbb{R}^n \to \mathbb{R}^n$ such that for all $a \in A$, $R_a$ is a Bregman function (Definition 3.7). Furthermore, $R_a$ is called contracting if $dR_a/da \leq 0$ for $a \in A$.

*Remark* 3.9. If $\alpha_t = 0$ for $t \geq T$ (for a $T > 0$), we recover a gradient flow with Riemannian metric $\left(\nabla_x^2 R_{a_T}\right)^{-1}$.

This implication is useful for proving our next result, which highlights under which conditions we can obtain convergence if we switch off regularization at some point during training. We will also verify later in our experiments that this is a promising dynamic regularization strategy. Our next Theorem 3.10 gives us the convergence we are looking for by using the newly defined contracting property above.

**Theorem 3.10.** *Consider the same setting as in Theorem 3.6. Furthermore, assume that $\alpha_t \geq 0$ and $\alpha_t = 0$ for all $t \geq T$, where $T > 0$. Moreover, for $a \in [b, 0]$, $R_a$ is a contracting Bregman function for some $b < 0$. Assume that for all $t \geq 0$ the integral $a_t := -\int_0^t \alpha_s ds \geq b$. For the loss function assume that $\nabla_x f$ is locally Lipschitz and $\text{argmin}\{f(x) : x \in \text{dom} R_{a_\infty}\}$ is non-empty. Then the following holds: If $f$ is quasi-convex, $x_t$ converges to a point $x_*$ which satisfies $\nabla_x f(x_*)^T (x - x_*) \geq 0$ for $x \in \text{dom} R_{a_\infty}$. Furthermore, if $f$ is convex, $x_*$ converges to a minimizer $f$ in the closed domain $\overline{\text{dom} R_{a_\infty}}$.*

Proof, see Theorem B.2 in the appendix. The proof consists of two parts: a) We show that the iterates are bounded up to time $T$ using the contracting property and quasi-convexity. b) We establish convergence after time $T$ using the geometric interpretation of the evolution of $x_t$.

**Optimality** To show optimality, we need more assumptions on the problem. As common in the context of mirror flows (Li et al., 2022; Jacobs & Burkholz, 2025), we recover under-determined linear regression, as follows. Let $\{(z_i, y_i)\}_{i=1}^n \subset \mathbb{R}^d \times \mathbb{R}$ be a dataset of size $n$. Given a reparameterization $g$ with regularization $h$, the output of the linear model on the $i$-th data is $z_i^T g(w)$. The goal is to solve

the regression for the target vector $Y = (y_1, y_2, \ldots, y_n)^T$ and input vector $Z = (z_1, z_2, \ldots, z_n)$.

**Theorem 3.11.** *Assume the same setting as Theorem 3.6. Furthermore, assume that $\alpha_t \geq 0$ and $\alpha_t = 0$ for all $t \geq T$, where $T > 0$. If $x_t$ converges when $t \to \infty$ and the limit $x_\infty = \lim_{t \to \infty} x_t$ satisfies $Z x_\infty = Y$, then the gradient flow minimizes the changed regularizer $R_{a_T}$:*

$$x_\infty = argmin_{x:Zx=Y} R_{a_T}(x). \tag{8}$$

Proof. See Theorem C.2.

This theorem extends known optimality results on matrix sensing (Gunasekar et al., 2017; Wu & Rebeschini, 2021) and diagonal linear networks (sparse coding) (Pesme et al., 2021; Woodworth et al., 2020; Jacobs & Burkholz, 2025).

### 3.4. Characterization of the Explicit Regularization

To make use of the established theoretical results in practice and develop promising regularization strategies, we characterize the explicit regularization $h$ for two important classes of reparameterizations: separable and quadratic reparameterizations.

**Separable Reparameterizations**   Our next result encompasses most previously studied reparameterizations within mirror flow framework(Woodworth et al., 2020; Pesme et al., 2021; Gunasekar et al., 2017).

**Corollary 3.12.** *Let $g$ be a separable reparameterization such that $g_i(w_i) = \sum_{j=1}^{m_i} g_{i,j}(w_{i,j})$ and $h(w) = \sum_{i=1}^{n} \sum_{j=1}^{m_i} h_{i,j}(w_{i,j})$, where $g_{i,j} : \mathbb{R} \to \mathbb{R}$ and $h_{i,j} : \mathbb{R} \to \mathbb{R}$. Furthermore, assume that $g$ and $h$ are analytical functions. Then, if and only if $h$ and $g$ satisfy*

$$h_{i,j} = c_{i,j} g_{i,j} \qquad \forall i \in [n], j \in [m_i],$$

*where $c_{i,j} \in \mathbb{R}$ is a constant, Theorem 3.6 applies.*

Proof. The result follows from the commuting relationship between $g$ and $h$. We use that the Wronskian between two analytical functions is zero if and only if they are linearly dependent (Bôcher, 1901).

The next two examples highlight the utility of Corollary 3.12.

*Example* 3.13. The reparameterization $g : \mathbb{R}^n \times \mathbb{R}^n \to \mathbb{R}^n$ such that $g(u, v) = u^2 - v^2$ with regularization of the form $h(u, v) = \sum_{i=1}^{n} c_u u_i^2 - c_v v_i^2$. Setting $c_u = 1$ and $c_v = -1$ leads to weight decay regularization on the reparameterization.

Example 3.13 has also been used to study the effect of stochasticity on overparameterized networks (Pesme et al., 2021). We present a more general class of examples that always results in a well-posed optimization problem, i.e., $h$ is positive.

*Example* 3.14. Consider the reparameterization $g : \mathbb{R}^n \times \mathbb{R}^n \to \mathbb{R}^n$ such that $g(u, v) = a(u) - b(v)$, where $a$ and $b$ are positive, analytical, increasing functions. Then, the regularization $\sum_{i=1}^{n} c_u a_i(u) - c_v b_i(v)$ can always be employed. By selecting $c_u \geq 0$ and $c_v \leq 0$, the optimization problem remains well-posed.

To give concrete examples, this approach encompasses $u^{2k} - v^{2k}$ (Woodworth et al., 2020) and $\log u - \log v$.

**Quadratic Reparameterizations**   Next, we will discuss the class of quadratic reparameterizations, as described in Theorem 4.16 in (Li et al., 2022).

**Theorem 3.15.** *In the setting of Theorem 3.10, consider the commuting quadratic parametrization $G : \mathbb{R}^D \to \mathbb{R}^d$ and $H : \mathbb{R}^D \to \mathbb{R}$, where each $G_i(w) = \frac{1}{2} w^T A_i w$ and $H(w) = \frac{1}{2} w^T B w$ with symmetric matrices $A_1, A_2, \ldots, A_d \in \mathbb{R}^{D \times D}$ and symmetric matrix $B \in \mathbb{R}^{D \times D}$ that commute with each other, i.e., $A_i A_j - A_j A_i = 0$ for all $i, j \in [d]$ and $B A_j - A_j B = 0$ for all $j \in [d]$. For any $w_{init} \in \mathbb{R}^D$, if $\nabla_w G_i(w_{init})_{i=1}^{d} = A_i w_{init} \,_{i=1}^{d}$ and $\nabla_w H(w_{init}) = B w_{init}$ are linearly independent, then the following holds:*

*1) $Q_a(\mu) = \frac{1}{4} || \exp(aB + \sum_{i=1}^{d} \mu_i A_i) w_{init} ||_{L_2}^2$ is a time-dependent Legendre function with domain $\mathbb{R}^d$.*

*2) For all $a \in \mathbb{R}$, $R_a$ is Bregman function with $dom R_a = \overline{range \nabla_x Q_a}$. Furthermore, if $B$ is positive semi-definite, then $\frac{dR_a}{da} \leq 0$, therefore Theorem 3.10 applies.*

Proof. The first statement is derived by applying Theorem 4.16 from (Li et al., 2022). The second statement follows from recognizing that $\exp(aB)$ acts as a linear transformation of the initialization $w_{init}$. Subsequently, applying Theorem 4.16 of (Li et al., 2022) gives the first part of the last statement. It remains to show that $R_a$ is contracting. Since $B$ is positive semi-definite, it follows that $\frac{d}{da} Q_a \geq 0$. By the reverse ordering property of convex conjugation, we have that $\frac{d}{da} R_a \leq 0$. For completeness, let $h > 0$, then for $a \in \mathbb{R}$, we have $Q_{a+h} \geq Q_a$. Applying the reverse ordering property implies $R_{a+h} \leq R_a$. Rearranging and dividing by $h$ gives $\frac{1}{h}(R_{a+h} - R_a) \leq 0$. Taking the limit $h \to 0$ concludes the proof.

*Remark* 3.16. For the time-dependent Bregman function in Theorem 3.15 to be contracting, $B$ needs to be positive semi-definite.

Theorem 3.15 encompasses recent works on the reparameterization $g(m, w) = m \odot w$, where $\odot$ denotes pointwise multiplication (Hadamard product). It has been proposed to sparsify neural networks (Jacobs & Burkholz, 2025), and extends work on matrix sensing and transformers (Wu & Rebeschini, 2021; Gunasekar et al., 2017; Sheen et al., 2024). Furthermore, $B = I$ corresponds to weight decay on the reparameterization, which is often used in practice.

Having identified classes where we can determine $h$, we next present applications to illustrate how time dependence influences the dynamics.

## 4. Analysis: Effects of Explicit Regularization

We introduce several time-dependent Legendre functions to demonstrate the wide applicability of our analysis. We aim to gain insights into how explicit regularization affects implicit bias during training. In particular, we focus on three distinct effects, as summarized below:

- Type of bias: The shape of $R_a$ changes with $a$.

- Positional bias: The global minimum of $R_a$ changes with $a$.

- Range shrinking: The range of $\nabla R_a$ can shrink due to a specific choice of $a$.

**The Reparameterization** $m \odot w$ The reparameterization $g(m, w) = m \odot w$ is an exemplary quadratic reparameterization, which can also be interpreted as the spectrum of more general quadratic reparameterizations. When the initialization satisfies $|w_0| < |m_0|$, then Theorem 3.15 holds. We can compute the time-dependent Bregman function $R_a(x)$:

$$\frac{1}{4} \sum_{i=1}^{d} x_i \text{arcsinh}\left(\frac{x_i}{A_i(a)}\right) - \sqrt{x_i^2 + A_i(a)^2} - x_i \log\left(\frac{u_{i,0}}{v_{i,0}}\right) \quad (9)$$

where $A_i(a) := 2\exp(2a)u_{i,0}v_{i,0}$ and $u_0 = (m_0 + w_0)/\sqrt{2}$ and $v_0 = (m_0 - w_0)/\sqrt{2}$. This adapts the hyperbolic entropy (Li et al., 2022; Woodworth et al., 2020; Wu & Rebeschini, 2021), which now is dependent on $a$. Note that we used Theorem 3.15 to find $R_a$, we can invert the corresponding function $Q_a(\mu)$, where $\mu = -\int_0^t \nabla_x f(x_s)ds$. The regularization thus affects the time-dependent Legendre function by changing $a$. This allows us to modulate between an implicit $L_2$ and $L_1$ regularization through explicit regularization (Jacobs & Burkholz, 2025). Moreover, $a$ also controls the location of the global minimum, a smaller $a$ corresponds to moving it closer to zero. Therefore, we both change the type of bias and the positional bias. Similarly, in case $w_0 = m_0 > 0$ we recover the entropy (Wu & Rebeschini, 2021):

$$\sum_{i=1}^{n} \left(\log\left(\frac{1}{B_i(a)}\right) - 1\right) x_i + x_i \log x_i, \quad (10)$$

where $B_i(a) := x_0 \exp(2a)$. Here $a$ modulates between maximizing and minimizing the $L_1$-norm. Note that both time-dependent Bregman functions are contracting on $a \in (-\infty, 0]$. Moreover, Figure 2 illustrates the effects of type of bias and positional bias for $m \odot w$.

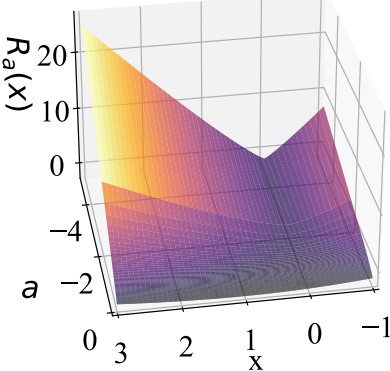

(a) Time-dependent hyperbolic entropy.

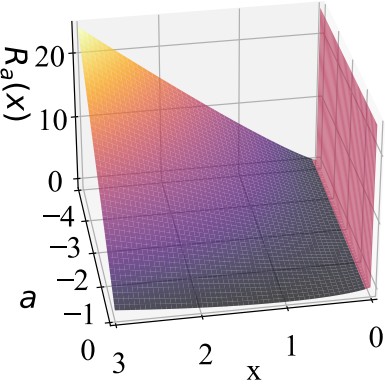

(b) Time-dependent entropy.

*Figure 2.* Illustrations of the positional bias and type of bias effects of explicit regularization on the time-dependent Legendre function. In both figures $a = -\int_0^t \alpha_s ds$. Depending on the initialization of $m \odot w$ the time-dependent Legendre function is given by Fig 2a or Fig 2b. Both exhibit a type change towards $L_1$ minimization.

**Quadratic Reparameterizations** Building on the characterization of the reparameterization $m \odot w$, we study the more general quadratic reparameterizations with weight decay ($B = I$). This covers multiple architectures, including matrix sensing, attention and LoRA, as explained in Table 1. For the dimensions of the parameters, see Table 6 in the Appendix. In general, however, the assumptions of Theorem 3.15 might not hold. In case of matrix sensing, we are able to apply both Theorem 3.15 and 3.11, where the time-dependent Bregman function for the eigenvalues is given by Eq. (10). Thus, weight decay modulates between maximizing and minimizing the nuclear norm of the matrix $X = UU^T$. The details are given in Appendix C. For attention, a common building block of Transformer architectures (Sheen et al., 2024; Tarzanagh et al., 2023), additional assumptions such as the alignment property are required. It is

*Table 1.* Quadratic parametrization.

| Matrix sensing | $UU^T$ |
|---|---|
| Attention | $SoftMax(QK^T)V$ |
| LoRA | $W_0 + AB$ |

worth noting that attention also has a value matrix $V$ and an activation function. Assuming $V$ is not trainable and that the function $f$ encompasses the activation function, the gradient flow dynamics of $X = QK^T$ is described by Theorem 3.6. This characterizes the implicit bias and can be interpreted as a proxy for training full attention. Similarly, for LoRA, a finetuning mechanism for LLMs (Hu et al., 2022; Wan et al., 2024), the training dynamics of $X = AB$ is described by Theorem 3.6, assuming the alignment property in addition.

As observed in (Khodak et al., 2021) for quadratic reparameterizations, weight decay promotes small nuclear norm. This is not the full picture, however. According to our results, weight decay changes the manifold geometry according to Eq. (7), which leads to an implicit bias that modulates between the Frobenius and nuclear norm of the matrix $X$. The eigenvalues in Eq. (9) are therefore subject to a time-dependent Bregman function $R_{a_t}$. This is the most accurate description of the implicit bias among the Bregman functions in Eq. (9) and Eq. (10) that we have discussed for the parameterization $m \odot w$, taking the initializations of both attention and LoRA into account. In case of attention, the matrices are randomly initialized, which makes a coupling between the spectrum of $K$ and $Q$ unlikely, i.e., $m_0 = w_0$. In case of LoRA, the initialization is $A = 0$ and $B$ is random. Regardless of the initialization, weight decay would still regularize towards the nuclear norm, but a coupled initialization would constrain the eigenvalues to be either positive or negative.

**The Reparameterization $u^{2k} - v^{2k}$**  The reparameterization $u^{2k} - v^{2k}$ serves as a proxy for deep neural networks (Woodworth et al., 2020) and provides an example of range shrinking due to explicit regularization. We consider the regularization $h(u,v) = \sum_{i=1}^{n} u_i^{2k} + v_i^{2k}$ as allowed by Corollary 3.12. The current reparameterization also exhibits a change in the implicit bias from $L_2$ to $L_1$, shown in Theorem 3 in (Woodworth et al., 2020). Unfortunately, there is no analytical formula available for the Legendre function in this case. Therefore we only derive the flow and its domain, which is the range of the time-dependent mirror flow. The flow $Q_{a_t}(\mu_t)$ is given by

$$d_k \left( \frac{1}{\mu_t + a_t + c_u} \right)^{\frac{2k}{2k-2}} - d_k \left( \frac{1}{-\mu_t + a_t + c_v} \right)^{\frac{2k}{2k-2}}, \quad (11)$$

where $d_k = ((2k-2)(2k))^{\frac{2k}{2k-2}}$, $\mu_t = -\int_0^t \nabla f(x_s)ds$ and $a_t = -\int_0^t \alpha_s ds$. We have that $\text{dom}\nabla_x R_a = \text{int dom} Q_a$

for $a$ fixed, where int refers to the interior of the domain (see Lemma 4.8 (Li et al., 2022)). Furthermore, note that $\mu \in (-c_u - a, c_v + a)$. Since $a_t$ is negative, the domain of $Q_a$ shrinks over time. Thus, the range of $\nabla_x R_a$ shrinks accordingly. This also lessens the set of acceptable solutions of the original optimization problem, which can make its solution harder. Figure 10 in Appendix F illustrates the effect of range shrinking with an approximation of the time-dependent Legendre function.

**Other Reparameterizations**  Appendix E and D present more reparameterizations. In particular, we analyze a reparameterization that induces an $L_1$ to $L_2$ change in the type of bias (in contrast to the more prevalent flipped change from $L_2$ to $L_1$). In addition, we highlight limitations of the framework by considering deeper reparameterizations.

## 5. Experiments

We conduct three experiments to support our theoretical analysis. The first experiment on matrix sensing illustrates the positional bias and type change following Theorem 3.11. Accordingly, we turn off weight decay at some point during training and compare it with a linear reparameterization with $L_1$ regularization. The second and third experiments are finetuning a pretrained transformer network and an LLM with LoRA, respectively. Both exhibit a gradual change of the implicit bias from Frobenius to nuclear norm, while demonstrating a lasting effect of dynamic regularization, which leads to better generalization. Notably, this highlights that our insights extend even to settings where our assumptions are not strictly met. Moreover, Appendix E discusses the range shrinking effect on sparse coding. Note that the change in positional bias is present in all experiments.

**Recovering the Sparse Ground Truth in Matrix Sensing**  We consider a matrix sensing experiment with the setup of (Wu & Rebeschini, 2021). Details can be found in Appendix C. The ground truth is a sparse matrix $X^*$ and the reparameterization is $X = UU^T$. When initialized with $U_0 U_0^T = \beta I$, the eigenvalues of $X$ satisfy Theorem 3.11 with the time-dependent Legendre function in Eq. (10). For experiments labeled "turn-off", we turn off the regularization at time $T = 625$. Figure 3 demonstrates that we recover the ground truth after turning off weight decay for the quadratic reparameterization. This can be explained by the fact that the positional bias of the eigenvalues moves closer to zero over time, which causes the type of bias to change to the nuclear norm (see Figure 6) in the appendix. Disabling weight decay in this context reveals the accumulated effects of regularization. In contrast, using constant weight decay would prevent us from exploiting this effect. Also note that a linear reparameterization with $L_1$ regularization could achieve high sparsity but at the expense of

reconstruction accuracy, as it cannot recover the ground truth. The geometry of the manifold that defines the implicit bias does not allow for it.

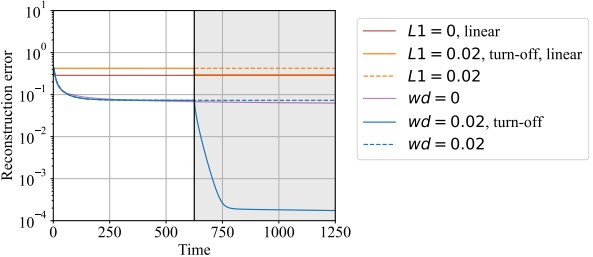

*Figure 3.* Recovering the sparse ground truth by turning weight decay off for matrix sensing at $T = 675$. In contrast, a linear reparameterization with $L_1$ regularization goes towards the minimal $L_2$ norm solution after a switch-off.

**Turning-off Weight Decay for LoRA and Attention**
Next, we aim to track the effect of the explicit regularization on the quadratic reparameterizations in Table 1. For LoRA, we calculate the nuclear norm and Frobenius norm of the matrix product $X = AB$, averaging these values across all layers, and then computing their ratio. For each attention layer, we apply the same procedure to the product of the query and key matrices $X = QK^T$. The ratio allows us to track the relative sparsity of the matrix. With LoRA, we fine-tune GPT2 (Radford et al., 2019) on the *tiny_shakespeare* (Karpathy, 2015) dataset, training for 500 iterations in two different types of settings. In case of "turn-off", we turn the weight decay off at iteration 200. Furthermore, we fine-tune a pretrained ViT on ImageNet for 300 epochs, turning the weight decay off at epoch 150. Figure. 4 shows that increasing weight decay reduces the reported norm ratio, indicating a change in the type of bias from $L_2$ to $L_1$. Moreover, when weight decay is turned off, the ratio intersects with other ratios that are attained by constant weight decay. This creates a "window of opportunity" for unconstrained training with a relatively low nuclear norm, leading to improved test accuracy (see Appendix F), in particular, in comparison with constant weight decay trajectories. For a ViT on ImageNet this can lead to more than $1\%$ improved validation accuracy for similar relative sparsity.

## 6. Discussion

We have introduced a framework for analyzing the impact of explicit regularization on implicit bias and provided a novel geometric interpretation of their interplay. By extending the mirror flow framework, we have outlined a method to control dynamic implicit bias through dynamic explicit regularization. Our analysis has characterized their joint effects on the training dynamics, including positional bias, type of bias, and range shrinking. Additionally, we have established a

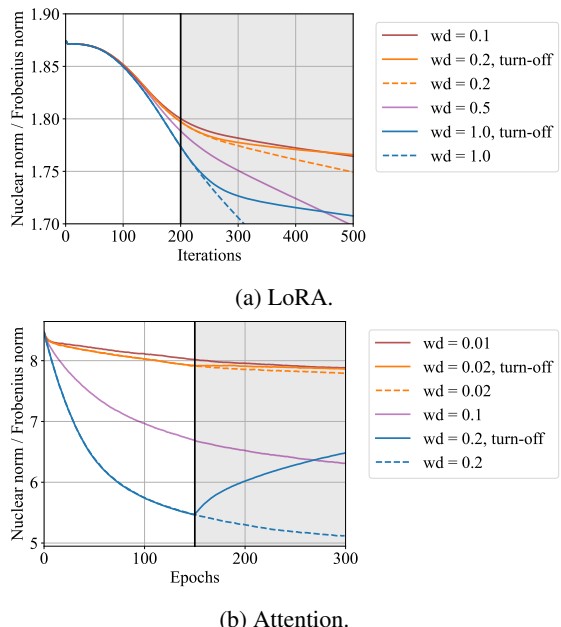

(a) LoRA.

(b) Attention.

*Figure 4.* Ratio between the nuclear norm and Frobenius norm for LoRA and attention. Training with higher weight decay and then turning it off in the shaded region allows for exploring the parameter space at higher sparsity. This creates a window opportunity to improve performance in a relatively sparse regime.

systematic procedure for identifying suitable regularizations for given reparameterizations and established convergence and optimality within our framework. As the implicit bias can change dynamically during training, it is associated with a time-dependent Legendre function, which might be conceptually of independent interest. To demonstrate the utility of our theory in applications, we have presented experiments on sparse coding, matrix sensing, attention in transformers, and LoRA fine-tuning. As we found, switching off weight decay at some point during training could improve generalization performance by exploiting the accumulated effect of past regularization. In future, our insights could guide the development of more effective regularization techniques that account for implicit bias, such as dynamic weight decay strategies tailored to specific model architectures. Moreover, our framework could be used to analyze the impact of early stopping and extended to incorporate other regularization factors like the impact of a large learning rate and stochastic noise.

## Acknowledgements

The authors thank Celia Rubio-Madrigal for proofreading and designing Figure 1. Moreover, the authors gratefully acknowledge the Gauss Centre for Supercomputing e.V. for funding this project by providing computing time on the GCS Supercomputer JUWELS at Jülich Supercomputing Centre (JSC). We also gratefully acknowledge funding from the European Research Council (ERC) under the Horizon Europe Framework Programme (HORIZON) for proposal number 101116395 SPARSE-ML.

## Impact Statement

This paper presents work whose goal is to advance the field of Machine Learning. There are many potential societal consequences of our work, none which we feel must be specifically highlighted here.

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

## A. Implicit bias framework

In this section, for completeness, we present the existing results for the mirror flow framework. Consider the optimization problem in Eq. (1) for a loss function $f : \mathbb{R}^n \to \mathbb{R}$

$$\min_{x \in \mathbb{R}^n} f(x).$$

We can use the implicit bias framework to study the effect of overparameterization. An overparameterization can be accomplished by introducing a function $g : M \to \mathbb{R}^n$, with $M$ a smooth manifold. For particular $g$, the reparameterization of the loss function $f$ leads to a mirror flow. A general framework is given in (Li et al., 2022) to study the implicit bias in terms of a mirror flow. Let $R : \mathbb{R}^n \to \mathbb{R}$ be a Legendre function (Definition 3.4), then the mirror flow is described by

$$d\nabla_x R(x_t) = -\nabla_x f(x_t)dt, \qquad x_{init} = g(w_{init}) \tag{12}$$

(Li et al., 2022) provide a sufficient condition for the reparameterization $g$ such that it induces a mirror flow Eq. (12). The Legendre function $R$ controls the implicit bias.

**Definition A.1.** (Legendre function Definition 3.8 ((Li et al., 2022))) Let $R : \mathbb{R}^d \to \mathbb{R} \cup \{\infty\}$ be a differentiable convex function. We say $R$ is a Legendre function when the following holds:

- $R$ is strictly convex on int(dom$R$).

- For any sequence $\{x_i\}_{i=1}^{\infty}$ going to the boundary of dom$R$, $\lim_{i \to \infty} ||\nabla R(x_i)||_{L_2}^2 = \infty$.

In order to recover the convergence result in Theorem 4.14 in (Li et al., 2022) the function $R$ also needs to be a Bregman function, which we define in Definition 3.7.

**Definition A.2.** (Bregman function Definition 4.1 (Alvarez et al., 2004)) A function $R$ is called a Bregman function if it satisfies the following properties:

- dom$R$ is closed. $R$ is strictly convex and continuous on dom$R$. $R$ is $C^1$ on int(dom$R$)).

- For any $x \in$ dom$R$ and $\gamma \in \mathbb{R}$, $\{y \in$ dom$R | D_R(x, y) \leq \gamma\}$ is bounded.

- For any $x \in$ dom$R$ and sequence $\{x_i\}_{i=1}^{\infty} \subset$ int(dom$R$) such that $\lim_{i \to \infty} x_i = x$, it holds that $\lim_{i \to \infty} D_R(x, x_i) \to 0$.

For a reparameterization to induce a mirror flow with a corresponding Legendre function we first have to give two definitions. Furthermore, we define $\partial g$ as the Jacobian of the function $g$.

**Definition A.3.** (Regular Parmeterization Definition 3.4 (Li et al., 2022)) Let $M$ be a smooth submanifold of $\mathbb{R}^D$. A regular parameterization $g : M \to \mathbb{R}^n$ is a $C^1$ parameterization such that $\partial G(w)$ is of rank $n$ for all $w \in M$.

For the second definition, we first need to define what a Lie bracket is.

**Definition A.4.** (Lie bracket Definition 3.4 (Li et al., 2022)) Let $M$ be a smooth submanifold of $\mathbb{R}^D$. Given two $C^1$ vector fields $X, Y$ on $M$, we define the Lie Bracket of $X$ and $Y$ as $[X, Y](w) := \partial Y(w)X(w) - \partial X(w)Y(w)$.

**Definition A.5.** (Commuting Parameterization Definition 4.1 (Li et al., 2022)) Let $M$ be a smooth submanifold of $\mathbb{R}^D$. A $C^2$ parameterization $g : M \to \mathbb{R}^d$ is commuting in a subset $S \subset M$ iff for any $i, j \in [n]$, the Lie bracket $\left[\nabla g_i, \nabla g_j\right](w) = 0$ for all $w \in S$. Moreover, we call $g$ a commuting parameterization if it is commuting in the entire $M$.

Besides these two definitions, we need to make an additional assumption on the flow of the solution. We define the solution of the gradient (descent) flow of a function $f : M \to \mathbb{R}^n$ initialized at $x \in M$

$$dx_t = -\nabla_x f(x_t)dt \qquad x_0 = x \tag{13}$$

as $x_t = \phi_x^t(x)$ which is well defined if the solution exists. Using this we can make the following assumption.

**Assumption A.6.** (Assumption 3.5 (Li et al., 2022)) Let $M$ be a smooth submanifold of $\mathbb{R}^D$ and $g : M \to \mathbb{R}^n$ be a reparameterization. We assume that for any $w \in M$ and $i \in [n]$, $\phi_{g_i}^t(w)$ is well-defined for $t \in (T_-, T_+)$ such that either $\lim_{t \to T_+} ||\phi_{g_i}^t(w)||_{L_2} = \infty$ or $T_+ = \infty$ and similarly for $T_-$. Also, we assume that for any $w \in M$ and $i, j \in [n]$, it holds that for $(t, s) \in \mathbb{R}^2$ that $\phi_{g_i}^s \circ \phi_{g_j}^t(w)$ is well-defined iff $\phi_{g_j}^t \circ \phi_{g_i}^s(w)$

Using these definitions we state the known result for mirror flow.

**Theorem A.7.** *(Theorem 4.9 (Li et al., 2022)) Let $M$ be a smooth submanifold of $\mathbb{R}^D$ and $g : M \to \mathbb{R}^n$ be a commuting and regular parameterization satisfying Assumption A.6. For any initialization $w_{init} \in M$, consider the gradient flow for any time-dependent loss function $L_t : \mathbb{R}^d \to \mathbb{R}$:*

$$dw_t = -\nabla_w L_t(g(w_t))dt, \qquad w_0 = w_{init}.$$

*Define $x_t = g(w_t)$ for all $t \geq 0$, then the dynamics of $x_t$ is a mirror flow with respect to the Legendre function $R$ given by Lemma 4.8 in (Li et al., 2022), i.e.,*

$$d\nabla_x R(x_t) = -\nabla_x L_t(x_t)dt, \qquad x_0 = g(w_{init}).$$

*Moreover, this $R$ only depends on the initialization $w_{init}$ and the reparameterization $g$, and is independent of the loss function $L_t$.*

We have used Theorem A.7 to show the Theorem 3.6 in the main text. Moreover, we recover the convergence result for Bregman functions in Theorem 3.10. The details of these results are presented in Appendix B.

## B. Proofs of Section 3

**Theorem B.1.** *Let $g : M \to \mathbb{R}^n$ and $h : M \to \mathbb{R}$ be regular and commuting parameterizations satisfying Assumption A.6. Then there exists a time-dependent Legendre function $R_a$ such that*

$$d\nabla_x R_{a_t}(x_t) = -\nabla_x f(x_t)dt, \qquad x_0 = g(w_{init})$$

*where $a_t = -\int_0^t \alpha_s ds$. Moreover, $R_a$ only depends on the initialization $w_{init}$ and the reparameterization $g$ and $h$ and is independent of the loss function $f$.*

Proof. Consider the time-dependent loss function $L_t(x, y) = f(x) + \alpha_t y$. Applying Theorem A.7 implies there is a Legendre function $R(x, y)$ such that

$$\begin{cases} \nabla_x R(x_t, y_t) = -\int_0^t \nabla_x f(x_s)ds \\ \partial_y R(x_t, y_t) = -\int_0^t \alpha_s ds. \end{cases} \tag{14}$$

We use Eq. (14) to derive the time-dependent Legendre function. First note that $\partial_y \partial_y R(x, y) > 0$ for $(x, y) \in domR$ since $R$ is strictly convex. This implies that the map $y \to \partial_y R(x, y)$ is invertible. Let the inverse be denoted by $Q(x, a)$, where in the dynamics $a_t = -\int_0^t \alpha_s ds$. Plugging $Q$ into the first part of Eq. (14) gives us

$$\nabla_x R\left(x_t, Q\left(x_t, a_t\right)\right) = -\int_0^t \nabla_x f\left(x_s\right) ds, \tag{15}$$

where $\nabla_x$ is still the derivative with respect to the first entry. Eq. (15) looks already like a time-dependent mirror flow. We show now that there exists a Legendre function $R_\alpha$ with the map $\nabla_x R\left(x, Q\left(x, \alpha\right)\right)$ as the gradient. This we can do by showing that the Hessian is symmetric and positive definite and that the $R_\alpha$ is essentially smooth.

By implicitly differentiating, we make the following observation:

$$\frac{dQ}{dx} = -\frac{1}{\partial_y \partial_y R(x, Q)} \nabla_x \partial_y R(x, Q).$$

Next, we compute the Hessian and apply observation B:

$$\begin{aligned} \nabla_x^2 R_\alpha &= \nabla_x^2 R(x, Q) + \partial_y \nabla_x R(x, Q) \cdot \frac{dQ}{dx} \\ &= \nabla_x^2 R(x, Q) - \frac{1}{\partial_y \partial_y R(x, Q)} \partial_y \nabla_x R(x, Q) \nabla_x \partial_y R(x, Q)^T. \end{aligned}$$

Observe that this matrix is symmetric as it is a sum of symmetric matrices. It remains to be shown that the Hessian matrix is positive definite. For this, we use that $\nabla_x^2 R$ and $\partial_y^2 R$ are positive definite. This implies that the the Hessian of $R$ inverse is PD. We use the block inversion matrix formula for matrix $M$,

$$M = \begin{bmatrix} A & B \\ C & D \end{bmatrix}$$

where $A$ and $D$ are square and invertible. For notation clarity, define the Schur complement of $D$ as:

$$S = A - BD^{-1}C$$

Then the inverse of $M$ is given by:

$$M^{-1} = \begin{bmatrix} S^{-1} & -S^{-1}BD^{-1} \\ -D^{-1}CS^{-1} & D^{-1} + D^{-1}CS^{-1}BD^{-1} \end{bmatrix}$$

The first block entry of the matrix $\nabla^2 R$ is given by the inverse Shur complement:

$$\left(\nabla_x^2 R(x, y) - \frac{1}{\partial_y \partial_y R(x, y)} \partial_y \nabla_x R(x, y) \nabla_x \partial_y R(x, y)^T\right)^{-1}$$

which is also PD. Now this implies the result as the inverse of $\nabla_x^2 R_\alpha$ is PD. It follows that there exists a function $R_a$ such that $\nabla R_a = \nabla_x R(x, Q(x, a))$ by Corollary 16.27 in (Lee, 2013), concluding the first part.

Finally, $R_a$ is essentially smooth by construction, using that $R$ is essentially smooth. The boundary $bn(R_a)$ by construction is the set of points $x^*$ that have a sequence $x_n \in domint\nabla_x R(\cdot, Q(\cdot, a))$ such that if $x_n \to x^*$ we have $||\nabla R|| \to \infty$. Suppose that $R_a$ is not essentially smooth then there exists a sequence $\{x_n\}$ with $x_n \to bd(R_a)$ as $n \to \infty$ such that $\lim_{n\to\infty} ||\nabla_x R(x_n, Q(x_n, a))||_{L_2}^2 < \infty$. Nevertheless, $R$ is essentially smooth this implies that

$$\lim_{n\to\infty} ||\nabla_y R(x_n, Q(x_n, a))||^2 = a^2 = \infty,$$

leading to a contradiction. Hence, $R_a$ is a Legendre function with the domain similarly constructed as the boundary. $\square$

**Theorem B.2.** *Assume the same settings as Theorem 3.6. Furthermore assume that for $\alpha_t \geq 0$ there is a $T > 0$ such that for $t \geq T$, $\alpha_t = 0$. Moreover for $a \in [b, 0]$, $R_a$ is a contracting Bregman function for some $b < 0$. Assume that for all $t \geq 0$ the integral $a_t := -\int_0^t \alpha_s ds \geq b$. For the loss function assume that $\nabla_x f$ is locally Lipschitz and $argmin\{f(x) : x \in domR_{a_\infty}\}$ is non-empty. Then if $f$ is quasi-convex $x_t$ converges to a point $x_*$ which satisfies $\nabla_x f(x_*)^T (x - x_*) \geq 0$ for $x \in domR_{a_\infty}$. In addition if $f$ is convex $x_*$ converges to a minimizer $f$ in $\overline{domR_{a_\infty}}$.*

Proof. We can bound the trajectory of $x_t$ by using the time-dependent Bregman divergence. The divergence between a critical point $x^*$ of $f$ and the iterates $x_t$ is given by

$$D_{a_t}(x^*, x_t) := R_{a_t}(x^*) - R_{a_t}(x_t) - \nabla_x R_{a_t}(x_t)^T (x^* - x_t) \geq 0$$

Note that the contracting property implies that for $a_2 \leq a_1$ we have $domR_{a_2} \subset domR_{a_1}$. Thus, a critical point $x^*$ in $domR_{a_\infty}$ is in $domR_{a_t}$. Hence, the divergence is well-defined. Due to that $f$ is quasi convex and $R_a$ contracting we have that $D_{a_t}(x^*, x_t)$ is bounded. From the contracting property it follows that $R_{a_\infty}(x^*) \geq R_{a_t}(x^*)$. By definition of a Bregman function, we have that $x_t$ stays bounded for all $t \geq 0$. It follows that $x_T$ is in the domain of $R_{a_\infty}$ and bounded. Therefore, we have that $D_{a_t}(x^*, x_t) \leq R_{a_\infty}(x^*) - R_{a_t}(x_t) - \nabla_x R_{a_t}(x_t)^T (x^* - x_t) =: W_t$. Now we show that the evolution of $W_t$ is decaying, implying that $D_{a_t}(x^*, x_t)$ is bounded. The evolution is given by

$$\begin{aligned} dW_t &= \alpha_t \frac{d}{da_t} R_{a_t}(x_t) dt - \nabla_x R_{a_t}(x_t) dx_t + \nabla_x R_{a_t}(x_t) dx_t - d\nabla_x R_{a_t}(x_t)^T (x^* - x_t) \\ &\leq -d\nabla_x R_{a_t}(x_t)^T (x^* - x_t) \\ &= +d\nabla_x f(x_t)^T (x^* - x_t) \\ &\leq 0 \end{aligned}$$

where we used that $\alpha_t \geq 0$ and the contracting property for the first inequality, the time dependent mirror flow relationship in the second and quasi-convexity for the last. Therefore $x_t$ stays bounded for $t \in [0, T]$. Now, using the geometeric interpretation Eq. (7) we have that the evolution of $\tilde{x}_t = x_{T+t}$ is a gradient flow on a Riemannian manifold with metric $\left(\nabla_x^2 R_{a_\infty}\right)^{-1}$. Therefore Theorem 4.14 in (Li et al., 2022) applies, which concludes the result. $\square$

## C. Optimality characterizing the implicit bias.

In this section, we state a general result for underdetermined linear regression extending Theorem 4.17 (Li et al., 2022). Moreover, we state a detailed result for matrix sensing extending Corollary 6 (Wu & Rebeschini, 2021).

For underdetermined linear regression, let $\{(z_i, y_i)\}_{i=1}^n \subset \mathbb{R}^d \times \mathbb{R}$ be a dataset of size $n$. Given a reparameterization $g$ with regularization $h$, the output of the linear model on the $i$-th data is $z_i^T g(w)$. The goal is to solve the regression for the label vector $Y = (y_1, y_2, \ldots, y_n)^T$. For notational convenience, we define $Z = (z_1, z_2, \ldots, z_n) \in \mathbb{R}^{n \times n}$.

In order to show optimality for underdetermined linear regression we use the following Lemma:

**Lemma C.1.** *(Lemma B.1 (Li et al., 2022)) For any convex function $R : \mathbb{R}^d \to R \cup \{\infty\}$ and $Z \in \mathbb{R}^{n \times d}$, suppose $\nabla R(x^*) = Z^T \lambda$ for some $\lambda \in \mathbb{R}^n$, then*

$$R(x^*) = \min_{x:Z(x-x^*)=0} R(x).$$

We now denote the function to be optimized by $f(x) = \tilde{f}(Zx - Y)$, to emphasize the linearity of the optimization problem.

**Theorem C.2.** *Assume the same settings as Theorem 3.6. Furthermore, assume that for $\alpha_t \geq 0$ there is a $T > 0$ such that for $t \geq T$, $\alpha_t = 0$. If $x_t$ converges as $t \to \infty$ and the convergence point $x_\infty = \lim_{t\to\infty} x_t$ satisfies $Zx_\infty = Y$, then*

$$x_\infty = argmin_{x:Zx=Y} R_{a_T}(x). \tag{16}$$

*Therefore, the gradient flow minimizes the changed regularizer $R_{a_T}$ among all potential solutions.* □

Proof. Since we assume convergence i.e. $Zw_\infty = y$, we have to show the KKT condition associated with Eq. (16) is satisfied. We have to show that $\nabla R_{a_T}(x^*) \in \text{span}\left(Z^T\right)$. This follows directly from integrating the time-dependent mirror flow for $t \geq T$:

$$\nabla R_{a_T}(x_t) - \nabla R_{a_0}(x_0) = -Z^T \int_0^t \nabla \tilde{f}\left(Zx_s - Y\right) ds \in \text{span}\left(Z^T\right).$$

Notice by definition of the Legendre function and its convex conjugate we have that $\nabla R_{a_0}(x_0) = 0$. Therefore, $\nabla R_{a_T}(x_t) \in \text{span}\left(Z^T\right)$, which further implies that $\nabla R_{a_T}(x_\infty) \in \text{span}\left(Z^T\right)$. Applying Lemma C.1 concludes the result. □

Theorem C.2 shows optimality for underdetermined linear regression. Moreover, together with Theorem 3.15 we extend results by a series of papers on quadratic reparameterizations (Jacobs & Burkholz, 2025; Li et al., 2022; Gunasekar et al., 2017; Azulay et al., 2021; Wu & Rebeschini, 2021).

We will focus now focus on one particular example: matrix sensing (Wu & Rebeschini, 2021; Gunasekar et al., 2017). The reason for this focus is to show the effect on the spectrum on the matrix. We show we can modulate between the Frobenius norm and nuclear norm similar to the modulation between $L_2$ and $L_1$ regularization as in (Jacobs & Burkholz, 2025). Moreover, we can induce a new grokking effect distinct from (Lyu et al., 2024; Liu et al., 2023), which considers large initialization and small weight decay.

Denote by $A_i \in \mathbb{R}^{n \times n}$ with $i \in [m]$ the sensing matrices and consider the loss function $f(X) = \frac{1}{2m} \sum_{i=1}^m \left(\langle A_i, X \rangle - y_i\right)^2$. Moreover, let $\mathbb{S}_n^+$ be the class of symmetric positive semi-definite matrices of size $n \times n$.

**Corollary C.3.** *Assume that the sensing matrices $A_i$'s are symmetric and commute, and that there exists a $X^* \in \mathbb{S}_n^+$ satisfying $f(X^*) = 0$. Moreover, assume that for $\alpha_t \geq 0$ there is a $T > 0$ such that for $t \geq T$, $\alpha_t = 0$. Then, the gradient flow defined by $\frac{dU_t}{dt} = -\nabla_X f(U_t U_t^T)U_t - \alpha_t U_t$ and any initialization satisfying $U_0 U_0^T = \beta I$ converges to a matrix $U_\infty$ minimizing*

$$\sum_{i=1}^n \left(log\left(\frac{1}{A_T}\right) - 1\right)\lambda_i + \lambda_i log\lambda_i \tag{17}$$

*among all global minima of f, where $\{\lambda_i\}_{i=1}^n$ denote the eigenvalues of the matrix $X_\infty := U_\infty U_\infty^T$ and $A_T := \beta \exp\left(-2 \int_0^T \alpha_s ds\right)$.*

Proof. Convergence follows from Theorem 3.15 and optimality follows from Theorem C.2. It is left to show that the corresponding time-dependent Bregman function $R_{a_t}(X_t)$ is given by Eq. (17). From the gradient flow we can derive the

time-dependent Bregman function:

$$R_{a_t}(X_t) = \text{Tr}\left(X_t\left(\log\left(\frac{1}{\beta\exp\left(-2\int_0^t \alpha_s ds\right)}\right) - 1\right) + X_t\log X_t\right).$$

Next, from the eigenvalue decomposition for symmetric matrices and the fact that all the above matrices are simultaneously diagonalizable, we get Eq. (17). $\square$

**Experiment** To illustrate the implication of Corollary C.3 we conduct an experiment on matrix sensing. The implication is that when we train with weight decay it is stored in the the time-dependent Bregman function. We can leverage this lasting effect by turning off the weight decay and reach the sparse (optimal) solution as in Eq. (16). This allows us to induce a grokking-like phenomenon.

We use a similar experimental setup as in (Wu & Rebeschini, 2021). Specifically, we generate a sparse groundtruth matrix $X^*$ by first sampling $U^* \in \mathbb{R}^{n\times r}$, where $r = 5$, with entries drawn i.i.d. from $N(0,1)$. We then set $X^* = U^*(U^*)^T$ and normalize it such that $||X^*||_{nuc} = 1$. We generate $m$ symmetric sensing matrices $A_i := \frac{1}{2}(B_i + B_i^T)$, where the $B_i$ entries are drawn i.i.d. from $N(0,1)$. We use learning rate $\eta = 0.25$, initialize $U_0 = I\beta$ with $\beta = 0.1$ and train for 5000 steps. We consider 3 scenarios:

- Train without weight decay i.e. $\alpha = 0$

- Train with weight decay $\alpha = 0.01, 0.02$

- Train with weight decay $\alpha = 0.02$ for 2500 steps and after that turn weight decay off, i.e. $\alpha = 0$.

The second scenario ($\alpha = 0.01$) and the third scenario are constructed such that the same amount of total regularization is applied at the end of training. Moreover, the same setup is used for the linear paramterization with $L_1$ regularization.

In Figure. 5 we present the evolution of the training and reconstruction error $||X^* - X_t||_{fro}^2$. We observe that in all scenarios the training error is below $10^{-2}$. In contrast, only the reconstruction error for the third scenario where we turn off the weight decay goes below $10^{-2}$. Therefore, by accessing the stored weight decay we can reach closer to the sparse solution illustrating Corollary C.3. This is also confirmed by the evolution of the nuclear norm in Figure. 6.

Note that keeping the weight decay on in the second scenario ($wd = 0.1$) also leads to a better reconstruction error than the overfitting of scenario one ($wd = 0$). Nevertheless, due to the explicit trade-off optimality is not possible. This is not in contradiction with the analysis in (Lyu et al., 2024), where very small weight decay is used. In the case of small weight decay, the trade-off is negligible.

The dynamics in Figure. 7 are similar to the dynamics of the grokking phenomenon: generalization happens later after no progress. Nevertheless, a key difference is that we induce it by turning off the weight decay in contrast to (Liu et al., 2023; Lyu et al., 2024). This leads to relatively "fast-grokking".

**Ablation with different schedules** Consider the family of schedules with constant regularization strength up to a specific time $T_i$ such that $\alpha_t = \alpha_0$ for $t \leq T_i$ and $\alpha_t = 0$ afterward, for $\alpha_0$ a constant. We choose $\alpha_0 = 0.02$ and $T_i = T/2$. In addition, we consider a linear and cosine decay schedule for the regularization with the same total strength (i.e., same integral), but the regularization is switched off after half of the training time to ensure convergence. To compare with the effect of turning off (t-o) the regularization, we include the constant schedule with regularization strength $\alpha_0 = 0.01$.

We observe in Table 2 that all schedules with decay or turn-off (t-o) converge to a solution with the same nuclear norm of the ground truth, confirming Theorem 3.6, while the constant schedule does not reach the ground truth.

*Table 2.* Performance of different regularization schedules.

| Schedule | Nuclear norm | Train loss | Rec error | Time to $10^{-7}$ train loss |
|---|---|---|---|---|
| Constant 0.01, no t-o | 0.93 | $7.2 \times 10^{-4}$ | $3.9 \times 10^{-2}$ | - |
| Linear decay | 1.00 | $1.8 \times 10^{-8}$ | $2.3 \times 10^{-4}$ | 661 |
| Cosine decay | 1.00 | $1.7 \times 10^{-8}$ | $2.1 \times 10^{-4}$ | 624 |
| Constant 0.02, t-o | 1.00 | $1.1 \times 10^{-8}$ | $1.7 \times 10^{-4}$ | 716 |
| Constant 0.2, t-o | 1.00 | $2.7 \times 10^{-10}$ | $2.7 \times 10^{-5}$ | 209 |
| Constant 2, t-o | 1.00 | $2.1 \times 10^{-10}$ | $2.4 \times 10^{-5}$ | 209 |
| Constant 20, t-o | 1.00 | $7.9 \times 10^{-13}$ | $1.4 \times 10^{-6}$ | 239 |

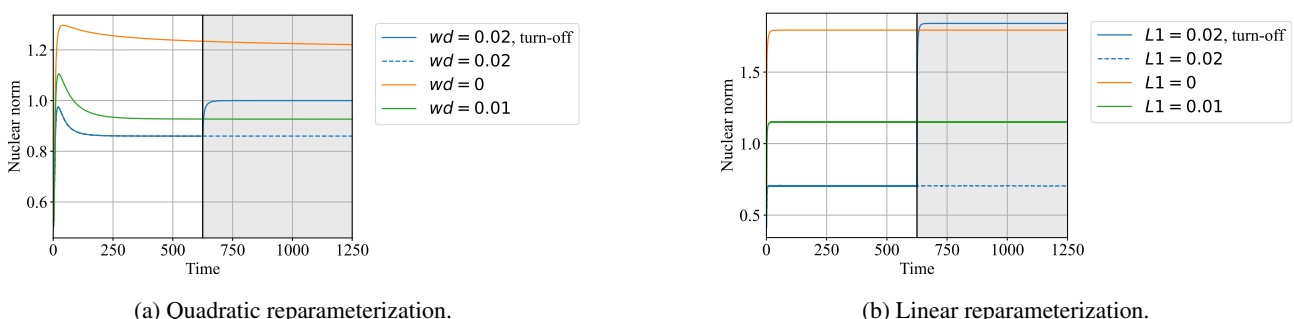

(a) Training error.

(b) Reconstruction error.

*Figure 5.* Train and reconstruction error for the matrix sensing experiment for quadratic parameterizations. Observe that both the training and reconstruction error decrease when the weight decay is turned off recovering the sparse ground truth.

(a) Quadratic reparameterization.

(b) Linear reparameterization.

*Figure 6.* Tracking the nuclear norm for both quadratic and linear reparameterization. In the case of the quadratic parameterization, the effect of the regularization gets stored resulting in a better reconstruction error in Figure. 3, whereas this does not happen in the linear case. In the linear case, the nuclear norm increases to the level of that without explicit $L_1$-regularization.

## D. Deeper reparameterization

In this section, we explore several reparameterizations and limitations of the framework. We show that Theorem 3.6 does not apply to linear parametrization. Moreover, Theorem 3.6 does not apply to overparameterizations with a depth larger than 2 and weight decay. Nevertheless, we show in experiments that similar effects can occur. We illustrate both the type change and range shrinking effect.

**Linear parametrization** From Corollary 3.12, we derive another corollary for non-overparameterized parametrization.

**Corollary D.1.** *Let $g(x) = x$ be the identity parametrization and $h \in C^2(\mathbb{R}^n, \mathbb{R})$. Then Theorem 3.6 applies if and only if, $h$ is given by $h(x) = \sum_{i=1}^{n} c_i x_i + d$ where $c_i, d \in \mathbb{R}$ are arbitrary coefficients.*

Proof. To apply the theorem, $h$ needs to be commuting with $g$, implying that $\partial_i \partial_i h = 0 \; \forall i \in [n]$, concluding the result. □

Corollary D.1 poses a limitation in the applicability of Theorem 3.6. Since $h$ is not positive for all $x \in \mathbb{R}^n$, the resulting optimization problem is ill-posed. Therefore, standard non-reparameterized loss functions cannot be analyzed in this manner.

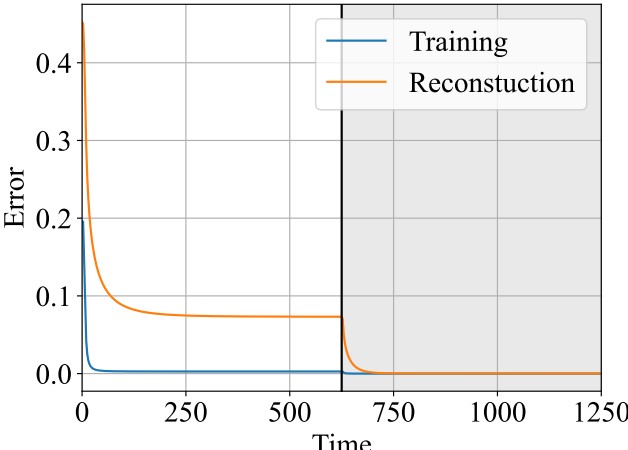

*Figure 7.* Training and reconstruction error for $\alpha = 0.02$ and turn-off. We recover the sparse ground truth by turning weight decay off for matrix sensing.

**Beyond quadratic reparameterization**   We show that the current framework excludes higher-order reparameterization with weight decay. In order to show that

**Theorem D.2.** *Let* $g : \mathbb{R}^k \to \mathbb{R}$ *be given by* $g(w) := \Pi_{i=1}^{k} w_i$ *, a* $k > 2$ *depth reparamterization. Moreover, let* $h : \mathbb{R}^k \to \mathbb{R}$ *and* $h(w) = \sum_{i=1}^{k} w_i^2$. *Then* $g$ *and* $h$ *do not commute.*

Proof. This follows directly from checking the commuting condition between $g$ and $h$:

$$[\nabla_w g, \nabla_w h](w) = \nabla_w g(w)\nabla_w^2 h(w) - \nabla_w h(w)\nabla_w^2 g(w)$$

$$= \begin{bmatrix} (4 - 2k)\, \Pi_{i \in [k] \setminus \{1\}} w_i \\ \vdots \\ (4 - 2k)\, \Pi_{i \in [k] \setminus \{k\}} w_i \end{bmatrix}.$$

In order for this to be equal to zero all products need to be zero. This implies that the gradient flow given by

$$dw_t = - \begin{bmatrix} \Pi_{i \in [k] \setminus \{1\}} w_{i,t} \\ \vdots \\ \Pi_{i \in [k] \setminus \{k\}} w_{i,t} \end{bmatrix} \odot \nabla_x f(g(w_t)) - \alpha_t w_t dt,$$

becomes $dw_t = -\alpha_t w_t dt$ and is independent of $f$. Hence, $g$ and $h$ do not commute $\square$

Theorem D.2 implies that we can not apply Theorem 3.6. We note that the commuting condition is only a sufficient criterion such that a pair $(g, h)$ is a time-dependent mirror flow.

**Experiment**   Although our theoretical result does not hold for reparametrizations with higher depth, we illustrate that the expected effects do occur as well for higher depth. We consider the reparameterization $m \odot w \odot v$ for diagonal linear networks and compare with $m \odot w$, both with weight decay. Moreover, we compare with the reparameterization $m$ with $L_1$ regularization to motivate the importance of the geometry, which is controlled by the time-dependent Legendre function. This is similar to the matrix sensing case explored in the previous section and main paper.

Let $d = 40$ be the amount of data points and $n = 100$ the dimension of the data. We generate independent data $Z_k \sim N(0, I_n)$ for $k \in [d]$. We assume a sparse ground truth $x^*$ such that $||x^*||_{L_0} = 5$. The training labels are generated by $y_k = Z_k^T x^*$. Moreover, the mean squared error loss function is used. The learning rate $\eta = 10^{-3}$ and we use weight decay $\alpha \in \{0.01, 0.1, 1\}$. We run the 100000 steps with weight decay, after that we run the same amount of steps without weight decay. We initialize $m = \mathbf{0}$ and $w = z = \mathbf{1}$, this ensures that both parametrizations are initialized at zero and have the same scaling. In this setup, we illustrate the type change similarly predicted for the parametrization $m \odot w$. Moreover,

we illustrate the range shrinking which occurs for higher depth parametrization $u^{2k} - v^{2k}$. Note that the ground truth has the following ratio between the $L_1$ and $L_2$ norm 2.23.

In Figure.8a we observe for $m$ that higher weight decay does not get closer to the ground truth after turning the $L_1$ regularization off. This is in line with the fact that the regularization is not stored in the geometry as described by Eq. (7). By turning off the regularization we converge to the closest solution in $L_2$ norm. This is best seen in Figure.9a, where the ratio increases above the value of the ground truth.

In Figure.8b we observe for $m \odot w$ that higher weight decay gets closer to the ground truth after turning the weight decay off. This is in line with the fact that the regularization is stored in the geometry as described by Eq. (7) and a type of bias change from $L_2$ to $L_1$. Furthermore, this is also confirmed in Figure.9b that for large weight decay, the ratio gets close to the ratio of the ground truth only after turning the weight decay off. This also illustrates Theorem 3.10 and 3.11.

In Figure.8c, we observe for the regularization strength $1e - 1$ a similar effect corresponding to the type of bias change from $L_2$ to $L_1$. In contrast, the higher regularization does not exhibit the same behaviour. We claim this is due to the range-shrinking effect. To motivate this is not due to the dynamics getting stuck at $x = \mathbf{0}$ we report the final value of the first parameter. The value is equal to $1.58$ which is not equal to either $0$ or the ground truth value $1$. To add to this, in Figure.9c we unveil that the ratio for large weight decay stays constant.

In conclusion, the type of bias can improve generalization, whereas $m \odot w$ even goes to the ground truth with high regularization, $m$ does not. Moreover, when we use higher order reparametrization such as $m \odot w \odot z$ we encounter a different phenomenon: range shrinking. To add to this, higher-order parametrization still exhibits the type of bias change in a certain range of regularization strength. Thus, our theoretical framework leads to verifiable predictions. These can be used to improve the training dynamics of neural networks in general.

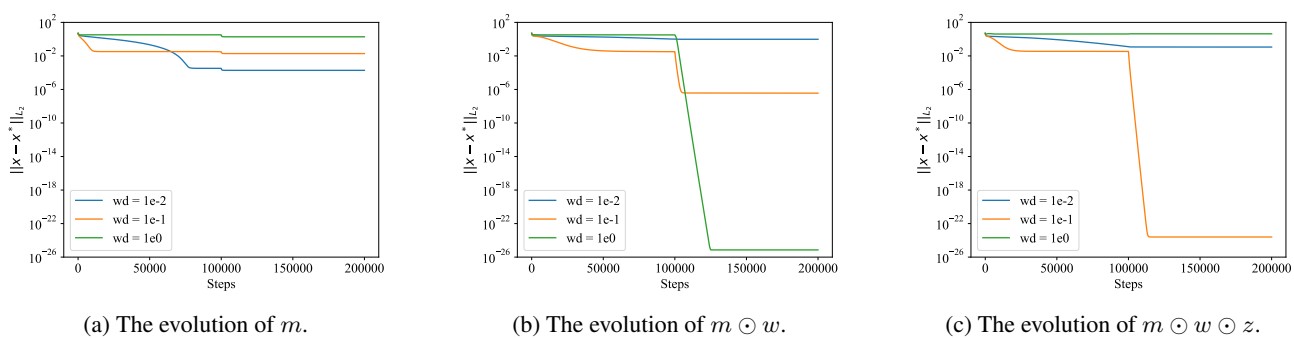

(a) The evolution of $m$.    (b) The evolution of $m \odot w$.    (c) The evolution of $m \odot w \odot z$.

*Figure 8.* Illustration of the effect of weight decay with higher order reparameterizations on generalization performance.

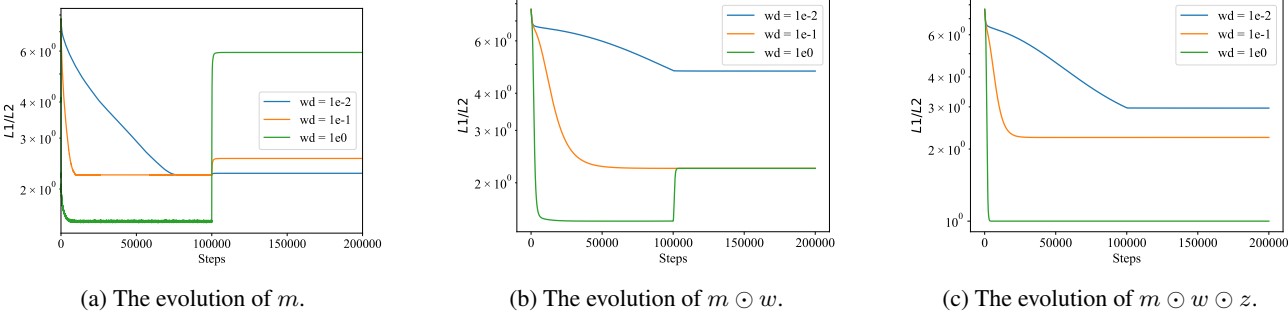

(a) The evolution of $m$.    (b) The evolution of $m \odot w$.    (c) The evolution of $m \odot w \odot z$.

*Figure 9.* The ratio between the $L_1$ and $L_2$ for diagonal linear networks.

# E. Sparse coding

We extend our study to the traditional sparse coding problem, with our proposed reparameterization substituting the standard sparse coding step. For this experiment, we use the Olivetti faces dataset. We denote the dictionary with $D$, labels with $z$, the code with $g(u, v)$ and regularization with $h(u, v)$. The feature dimension of $D$ is $n$. This is solved as a linear regression problem with mean squared error. We have used a learning rate $\eta = 0.001/Lip(D)$ where $Lip(D)$ denotes the resulting Lipschitz constant of the optimization problem depending on the dictionary $D$. In addition, we set the number of features $n = 50$ and run for 100 iterations.

**The reparameterization $u^{2k} - v^{2k}$**    In this context, we reparameterize the sparse code as $g(u, v) = u^{2k} - v^{2k}$ and set the regularization $h(u, v) = \sum_{i=1}^{n} u_i^{2k} + v_i^{2k}$ as discussed in the main paper. This parameterization exhibits range shrinking as illustrated by the time-dependent Legendre function in Figure 10.

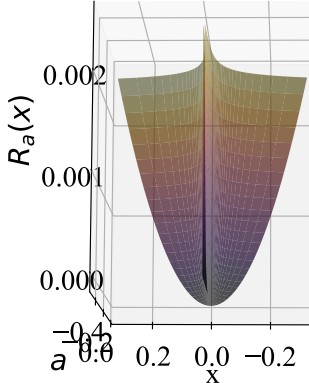

*Figure 10.* The evolution of the approximated $R_a$ associated with $u^4 - v^4$, where $a = -\int_0^t \alpha_s ds$.

The parameters are initialized as $u_0 = \frac{1}{2}(\sqrt{x^2 + \beta^2} + x)^{\frac{1}{2k}}$ and $v_0 = \frac{1}{2}(\sqrt{x^2 + \beta^2} - x)^{\frac{1}{2k}}$, where $\beta = 1$, $x \sim \mathcal{N}(0, I_n)$, and all operations are pointwise. We set the regularization strength to $\alpha = 0.001$ and explore various values of $k \in [7]$. Throughout the training process, we track two key metrics: the reconstruction mean squared error (MSE) and the nuclear norm of the sparse code, $g(u, v) = u^{2k} - v^{2k}$. The evolution of the nuclear norm is presented in Figure.11a. We observe the effect of the range shrinking for $k > 1$, for larger $k$ the evolution of the nuclear norm becomes stationary faster. This indicates that the range in which the time-dependent Legendre function is allowed to move has shrunk. The shrinking also causes the MSE to converge faster for large $k$, shown in Figure.11b.

**The reparameterization $\log(u) - \log(v)$**    We consider a novel reparameterization. In the main text, we have seen that the regularization changed the type of bias from $L_2$ to $L_1$. We now consider a reparameterization with explicit regularization that leads to the opposite type of bias change. The reparameterization is $g(w) = \log(u) - \log(v)$. The regularization found in Corollary 3.12 is $h(w) = \sum_{i=1}^{n} \log(u_i) + \log(v_i)$. Then for $u, v > 1$ we can apply Theorem 3.6.

We now give the resulting time-dependent Legendre function. The time-dependent Legendre function is

$$R_a(x) = \frac{1}{4} \sum_{i=1}^{n} \left(u_{0,i}^2 - 2a\right) \log\left(e^{-2x_i} + 1\right) + \left(v_{0,i}^2 - 2a\right) \log\left(e^{2x_i} + 1\right) \ \forall a < \frac{1}{2}\min\{u_{0,i}^2, v_{0,i}^2\}.$$

The global minimum is centered at $\nabla_x R_a = 0$ and is given by $\log\left(\sqrt{u_0^2 - 2a}\right) - \log\left(\sqrt{v_0^2 - 2a}\right)$. Thus a shift occurs when $a$ changes, illustrating the positional bias. Moreover, to illustrate the type change, consider the balanced initialization $u_0 = v_0 = \beta I$, the Legendre function is then given by

$$R_a(x) = \frac{1}{4}\left(\beta^2 - 2a\right) \sum_{i=1}^{n} \log\left(2\cosh(x_i)\right)$$

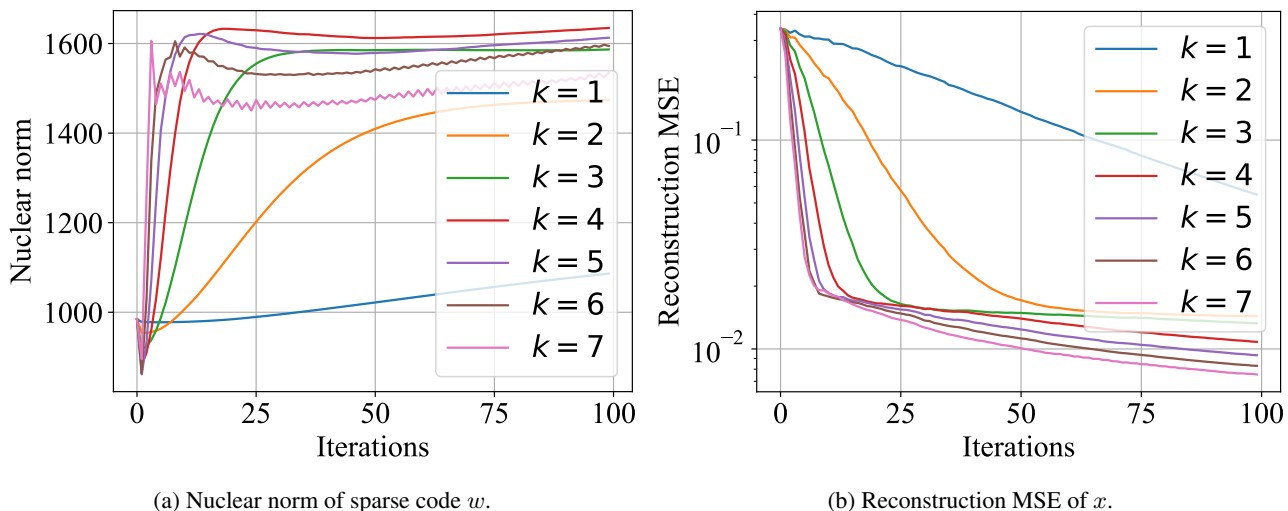

(a) Nuclear norm of sparse code $w$.

(b) Reconstruction MSE of $x$.

*Figure 11.* Results for sparse coding reparameterisation $g(w) = u^{2k} - v^{2k}$

which resembles the log-cosh loss function with vertical rescaling. The rescaling changes the type of bias from $L_1 \to L_2$. The type here is $L_2$ close to the origin and $L_1$ further away from zero. Due to the scaling, it becomes closer and closer to $L_2$. This is illustrated in Figure.12. Furthermore, we will show in experiments that the type change is crucial for generalization.

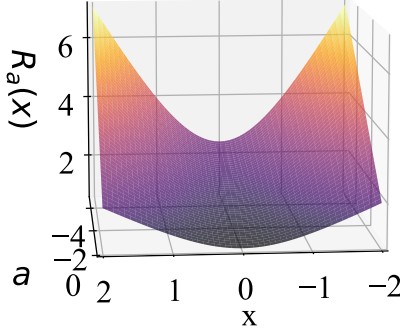

*Figure 12.* From $L_1$ to $L_2$ implicit bias, with $a = -\int_0^t \alpha_s ds$.

In this context, we reparameterize the sparse code as $g(w) = \log(u) - \log(v) \in \mathbb{R}^n$ and replace the regularization as discussed. We initialize the parameters as $u_0 = 1/(\beta(1 + e^{-x}))$ and $v_0 = 1/(\beta(1 + e^x))$, where $\beta = 1$ and $x = 0.1$. Note that the initialization is different for stability reasons. We explore various values for $\alpha \in \{0.0001, 0.001, 0.01, 0.1, 0.0, 1.0\}$. During the training process, we track two key metrics: the reconstruction Mean Squared Error (MSE) and the nuclear norm of the sparse code, defined as $g(w) = \log u - \log v$. The results are illustrated in Figure.13. We observe that higher regularization leads to a faster increase in the nuclear norm, which confirms the movement to $L_2$ regularization. This leads to a construction error.

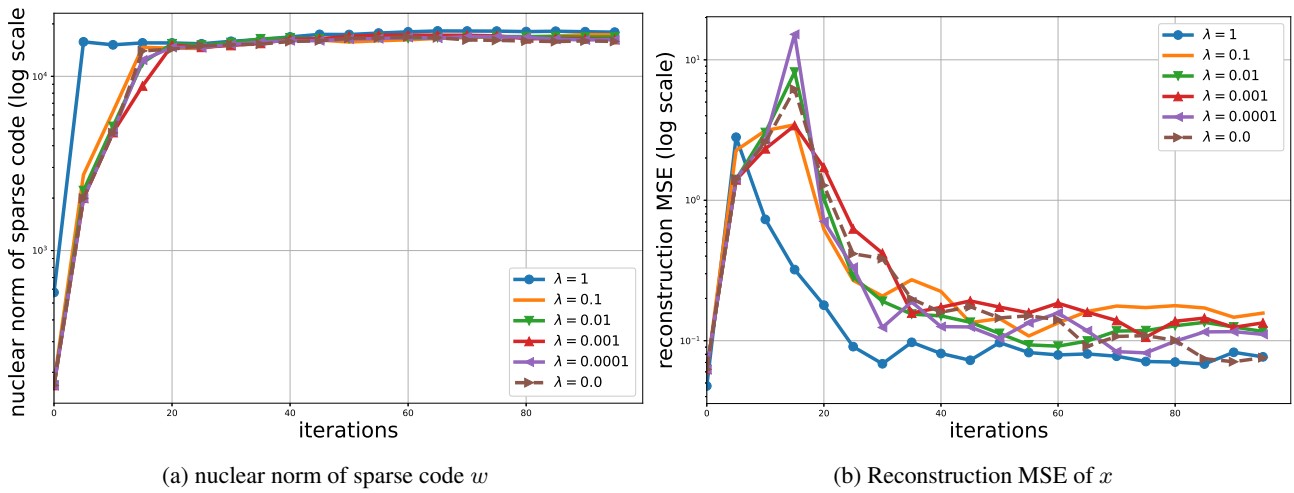

(a) nuclear norm of sparse code $w$

(b) Reconstruction MSE of $x$

*Figure 13.* Results for sparse coding reparameterisation $g(w) = \log u - \log v$

## F. Hyperparameters and additional figures on quadratic reparameterizations

We present the experimental details and additional figures such as validation error and accuracy. Moreover, we present the evolution of the time-dependent Legendre functions corresponding to $m \odot w$ in Figures.2a and 2b . In the case of attention, we used the optimizer AdamW with a learning rate $1e-4$ and a constant learning rate. We start training from a pretrained tiny-ViT on ImageNet. Moreover, the CIFAR10 experiment is run over 3 seeds. Finally, for LoRA we use SGD with momentum $(0.9)$, constant learning rate $2e-4$, LoRA rank $8$, alpha $8$ and no drop-out. The GPT-2 experiment is run over 2 seeds.

The validation accuracy is given in Table 4 for the attention experiment. The validation loss is given in Table 5 for the LoRA experiment. In all cases turning off the weight decay leads to an improved validation score (accuracy or error). In Table 3 we provide an ablation for various turn off points of the weight decay for strength $0.2$. Turning off leads to better generalization at the intersection point or even before intersection.

*Table 3.* Validation accuracy and ratio at different weight decay turn-off points.

| WD Off | Intersect | Val Acc (WD) | Val Acc (Off) | Norm Ratio |
|--------|-----------|--------------|---------------|------------|
| 50     | 104       | 70.7         | 72.4          | 6.9        |
| 100    | 195       | 70.2         | 72.6          | 6.5        |
| 150    | 270       | 71.1         | 73.4          | 6.4        |
| 200    | None      | 70.3 (6.3)   | 72.5 (6.1)    | -          |

Furthermore, we provide training of a tiny-ViT from scratch on CIFAR10 with varying weight decay in Figure. 14. The learning rate is $1e-3$ and we use cosine warmup. Observe that higher weight decay is necessary to keep the ratio down at the end of the training. This can improve the validation error significantly as observed for $wd = 0.1$.

*Table 4.* Validation accuracy for tiny-ViT experiment (attention).

| Dataset  | $wd = 0.2$ | $wd = 0.2$, turn-off | $wd = 0.1$ | $wd = 0.02$ | $wd = 0.02$, turn-off | $wd = 0.01$ |
|----------|------------|----------------------|------------|-------------|-----------------------|-------------|
| ImageNet | 71.08      | **73.66**            | 71.76      | 74.576      | **75.24**             | 75.05       |
| CIFAR10  | 52.38($\pm$0.16) | **56.7($\pm$(0.39)** | 54.95($\pm$0.36) | 57.05($\pm$0.27) | **57.78($\pm$0.32)** | 57.33($\pm$0.44) |

*Table 5.* Validation loss for LoRA experiment on Shakespeare dataset.

| Architecture | $wd = 1.0$ | $wd = 1.0$, turn-off | $wd = 0.5$ | $wd = 0.2$ | $wd = 0.2$, turn-off | $wd = 0.1$ |
|---|---|---|---|---|---|---|
| GPT2-xl | 2.99 | **2.96** | 2.97 | 2.96 | **2.95** | 2.96 |
| GPT2 | $3.44(\pm0.00)$ | **$3.42(\pm0.00)$** | $3.43(\pm0.00)$ | $3.43(\pm0.01)$ | **$3.41(\pm0.00)$** | $3.42(\pm0.01)$ |

*Table 6.* Quadratic reparameterization.

| Matrix sensing | $UU^T$ | $U \in \mathbb{R}^{n \times n}$ |
|---|---|---|
| Attention | $Softmax(QK^T)V$ | $Q, K, V \in \mathbb{R}^{n_q \times d}$ |
| LoRA | $W_0 + AB$ | $A \in \mathbb{R}^{n \times r}, B \in \mathbb{R}^{r \times n}, W_0 \in \mathbb{R}^{n \times n}$ where $r << n$ |

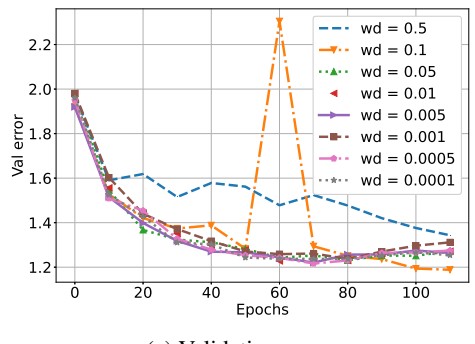

(a) Validation error.

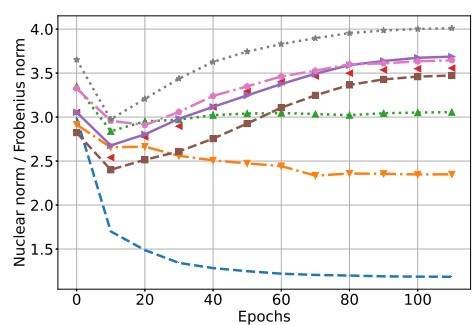

(b) Ratio of the nuclear norm and the Frobenius norm.

*Figure 14.* Varying the weight decay parameter for a ViT on CIFAR10. Higher weight decay leads to a lower ratio and can also lead to lower validation errors.

## G. Learning rate schedule

We further study the effect of the learning rate scheduler. Specifically, we run pre-trained ViT-tiny on ImageNet classification fine-tuning task. We set the learning rate to $1e-4$ with AdamW optimisers. We also vary the weight decay in the range $[0.001, 0.003, 0.005, 0.007, 0.01]$. Moreover, for each of the settings, we train two comparison experiments, one without a learning rate scheduler, and one with the popular CosineAnnealingWarmRestarts.

The results are shown in Figure.15. Furthermore, results with SGD optimizer are included in Figure.16. We observe in both figures that the validation accuracy increases for the decaying schedule in comparison to the constant schedule. Moreover, we again observe a decaying ratio, for stronger weight decay the ratio decreases more. Moreover, decaying the learning rate has a similar effect as turning off the weight decay on the implicit and explicit regularization. Note that the AdamW optimizer can have additional effects that also contribute to changing the ratio. Furthermore, regardless of the learning rate schedule, the ratio is decreasing indicating a modulation from Frobenius norm towards nuclear norm minimization.

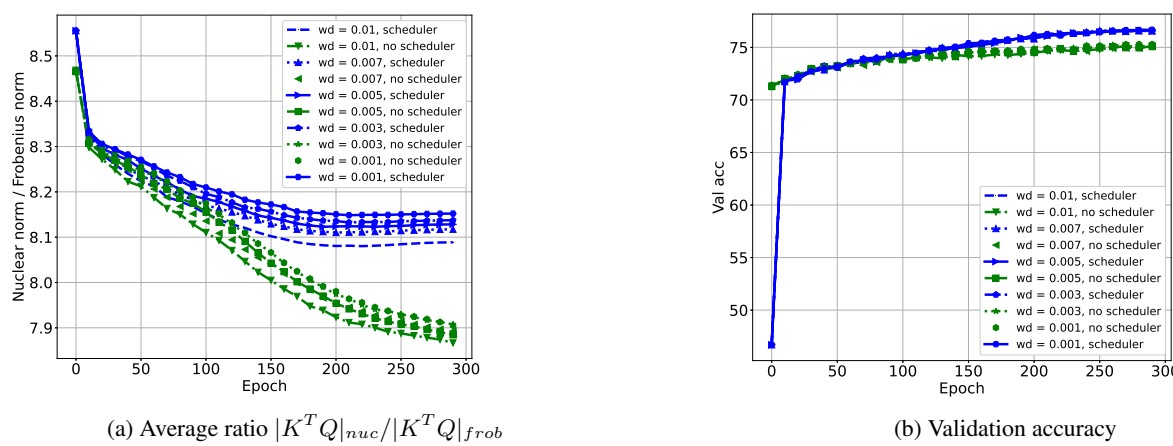

(a) Average ratio $|K^T Q|_{nuc}/|K^T Q|_{frob}$

(b) Validation accuracy

*Figure 15.* Results for ViT-tiny fine-tuning task with AdamW optimiser on ImageNet.

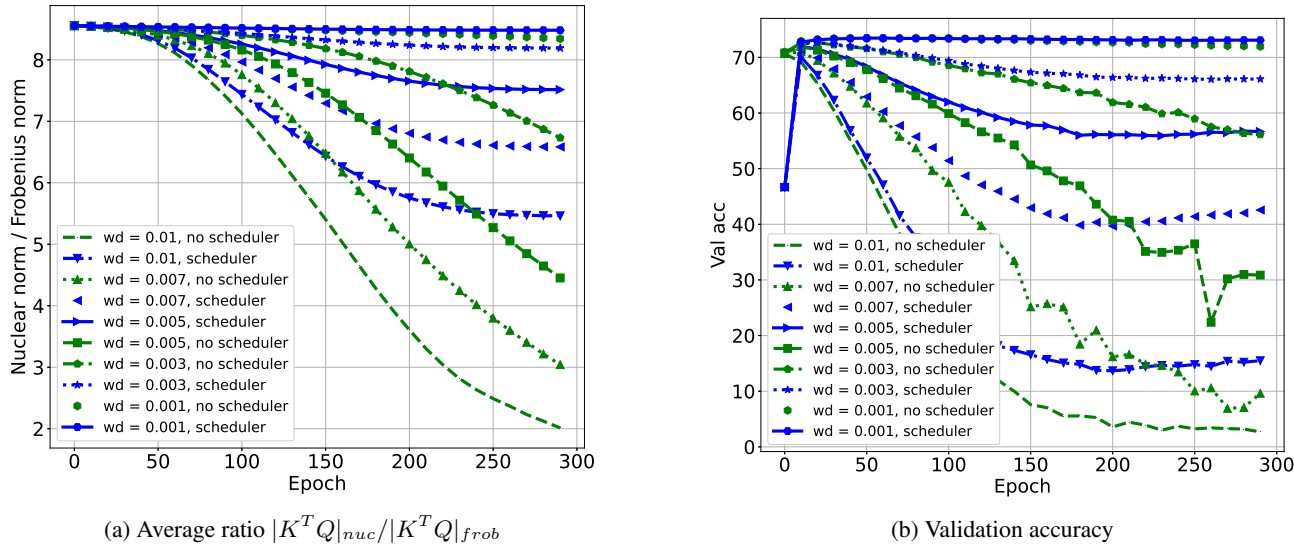

(a) Average ratio $|K^T Q|_{nuc}/|K^T Q|_{frob}$

(b) Validation accuracy

*Figure 16.* Results for ViT-tiny fine-tuning task with SGD optimiser on ImageNet.

