# OpenReview forum: "Mirror, Mirror of the Flow: How Does Regularization Shape Implicit Bias?"
_ICML.cc/2025/Conference — ICML 2025 poster_

### Official Review · Reviewer_PG36 · 2025-03-12

**Overall Recommendation:** 3

**Summary:**

The authors propose to analyze the effect of combining implicit and explicit regularization on training dynamics using the mirror flow framework. While their framework is more general, they specifically examine the impact of weight decay when it is turned off at a particular training step. To investigate this, the authors conduct several experiments to assess the influence of weight decay on both the sparsity of neural networks and their performance.

**Claims And Evidence:**

The authors suggest that their theoretical framework provides insights for designing appropriate regularization strategies, and they conduct experiments to support this claim. The author take the special case of turn-off weight decay to illustrate the application of their framework and let to future work do design temporal weight decay. In the experimental section, it would be beneficial to include an ablation study examining the impact of the time step at which weight decay is turned off. Additionally, deriving theoretical boundaries for (T,weight decay) and comparing them with the experimental results would strengthen the analysis.

**Essential References Not Discussed:**

I have no strong opinion on this matter, as I am not particularly specialized in this litterature.

**Experimental Designs Or Analyses:**

Regarding the **experimental section**, I appreciate that the authors demonstrate, in the case of linear hyperparameterization, that turning off weight decay improves the error in this particular setting, validating the theoretical results (Figure 2). However, I have some concerns about the experiments on real datasets in terms of accuracy and sparsity metrics:

1. **Tables 2 and 3 in the appendix:** The performance differences between the best turn-off setup and the best weight decay configuration (which is slightly smaller) do not appear to be very significant. In my view, it is difficult to draw a strong experimental conclusion because an additional hyperparameter (the step at which weight decay is turned off) needs to be tuned, and similar results can be achieved by selecting an appropriate weight decay value.

2. **Figure 3:** While the experiment illustrates the impact of the turn-off mechanism, the advantage of turning off weight decay in terms of the nuclear norm to Frobenius norm ratio is not entirely clear.

Could you clarify whether I have misinterpreted the analysis of these experiments?

**Methods And Evaluation Criteria:**

To validate their theoretical findings, the authors investigate a specific case where weight decay is switched off at a particular training step T. They analyze the impact on accuracy and the ratio of the Nuclear norm to the Frobenius norm across various architectures (e.g., LoRA, attention mechanisms) and datasets, including ImageNet and the Shakespeare dataset.

**Other Comments Or Suggestions:**

None

**Other Strengths And Weaknesses:**

Strengths: The paper is well-written, clear, and easy to understand.

Weaknesses: The proofs are less accessible at times due to brief explanations between certain steps, making it harder to follow the reasoning.

**Questions For Authors:**

I would appreciate responses to my theoretical and experimental analysis questions outlined in the "Theoretical Claims" and "Experimental Designs Or Analyses" sections.

**Relation To Broader Scientific Literature:**

I have no strong opinion on this matter, as I am not particularly specialized in this litterature.

**Theoretical Claims:**

I have reviewed the proofs of Theorems B.1 and B.2 and have a few questions:

1. **B.1 – L785:** I do not fully understand the justification for $\partial_y\partial_y R(x,y) > 0$. Given that the authors consider the loss function $ L(x,y) = f(x) + \alpha y $, the second derivative should be $\partial_y\partial_y R(x,y) = 0$. Could you clarify where my reasoning is incorrect?

2. **B.1 – L804:** I did not fully grasp your demonstration regarding why $\nabla^2_x R$ is positive definite. Could you provide additional explanation?

3. **B.2 – L843 to L846:** You state that  $d \nabla_x R^T_{a_t}(x^* - x_t) = - d \nabla_x f(x_t)^T (x^* - x_t).$  However, in my understanding, it should be  $d \nabla_x R^T_{a_t}(x^* - x_t) = - \nabla_x f (x^* - x_t).$ Consequently, I do not see why we can conclude that $\nabla_x f (x^* - x_t) \leq \nabla_x f (x_t)^T (x^* - x_t).$

---

> ### Author Rebuttal · Authors · 2025-03-31
>
> We appreciate the reviewer’s time and valuable feedback. We’re happy to address any further questions or concerns.
>
> **Turning-off at different points**
>
> We are happy to provide an ablation on turning off the regularization at different points for the vision transformers:
> We disable weight decay (WD) at different epochs during fine-tuning of Tiny-ViT on ImageNet. We train for 300 epochs with AdamW (lr=1e-4, WD=0.2) and disable WD at epochs [50, 100, 150, 200]. We report the intersection epoch of nuclear norm/Frobenius norm ratio and validation accuracy with/without WD off. If no intersection occurs, we report final validation accuracy and ratio. Results show better performance for similar ratios, supporting our theory. We will include this additional experiment in the revised manuscript.
>
> | WD Off | Intersect | Val Acc (WD) | Val Acc (Off) | Norm Ratio |
> |--------|----------|--------------|---------------|------------|
> | 50     | 104      | 70.7%        | 72.4%         | 6.9        |
> | 100    | 195      | 70.2%        | 72.6%         | 6.5        |
> | 150    | 270      | 71.1%        | 73.4%         | 6.4        |
> | 200    | None     | 70.3% (6.3)  | 72.5% (6.1)   | -          |
>
> **Theoretical bounds on $T$ and $\alpha$**
>
> For underdetermined linear regression (Thm 3.6), we search for the schedule with the fastest convergence given a fixed desirable $a_T = \int_0^T \alpha_s ds < \infty$. This is equivalent to regularizing as much as possible and turning off regularization quickly. Yet, it is known that the saddle point near the origin slows down the  training dynamics. Also, this is impacted by a finite learning rate inducing a discretization error. Table for Reviewer Eyxr identifies a solution to the expected trade-off.
> Yet, the optimal $T$ depends on intricacies of the loss landscape and the training dynamics. For example, for the goal of sparsifying a neural network, gradually moving the implicit bias from $L_2$ to $L_1$ works well [1]. Intuitively, we expect to train with regularization until we hit a loss basin to switch it off soon thereafter. A theoretical bound presents a great challenge that is beyond the scope of this paper with general framework character.
>
> [1] Jacobs, T. and Burkholz, R. Mask in the mirror: Implicit sparsification. In The Thirteenth International Conference on Learning Representations, 2025. URL https://openreview.net/forum?id=U47ymTS3ut.
>
>
>
> **Details of main proofs**
>
> $R(x,y)$ is unrelated to the loss function $L$. At the end of Theorem A.7, it is defined by the reparameterization, i.e. $(g,h)$, and the initialization. The second derivative is positive. This follows from the definition of a Legendre function (Definition A.1), which is strictly convex. We will refer to this explicitly.
> We can express the Hessian of $R$ in terms of block matrices that are derivatives with respect to $x$ and $y$ and then apply the block matrix inversion lemma (see  Proposition 2.8.7 in [1]). The first diagonal block matches exactly the inverse of the constructed Hessian for the time-dependent Legendre function. As the inverse of a positive definite matrix is positive definite, it follows from the definition of positive definiteness that also this diagonal block matrix is positive definite. To see this more precisely, denote the inverse of the Hessian matrix with $A$. $A$ is positive definite if and only if for all $z = (x,y) \in \mathbb{R}^{n+1}$, $z^T A z > 0$. Now restricting this to particular $z = (x,0)$ with $x \in \mathbb{R}^n$ yields that the diagonal block matrix that we are interested in is also positive definite.
> We suppressed the dependency of $\nabla R_{a_t}$ on $x_t$. We agree that it would be more clear to use $\nabla R_{a_t}(x_t)$ instead. In addition, we will separately explain the application of the time-dependent mirror flow relation and the contraction property to highlight how the contraction property is used.
> We will include these extra explanations in the revised manuscript. Moreover, we will provide the block matrix inversion calculation according to Proposition 2.8.7.
>
> [1] Bernstein, D. S. (2005). Matrix Mathematics: Theory, Facts, and Formulas with Application to Linear Systems Theory. http://engineering.nyu.edu/mechatronics/Control_Lab/ToPrint.pdf
>
> **Experiment interpretation**
>
> We agree that the improvement for the LORA experiment is lower, although significant. However, for the vision transformers, the increase in validation accuracy is substantial for the same ratio. The additional experiments above support this claim. Furthermore, we would like to emphasize that our experiment is not designed to maximize performance but to confirm the prediction of the theory. The desirability of a particular ratio is problem-dependent. For example, a lower rank (smaller ratio) might be desirable in cases where efficiency or interpretability is a priority. Moreover, the framework could be used to inspire and help design new regularization schedules that are tuned for better performance.

---

### Official Review · Reviewer_ZErA · 2025-03-13

**Overall Recommendation:** 3

**Summary:**

Motivated by the fact that the inductive bias of a trained neural network depends on implicit biases of the training algorithm and explicit regularization such as weight decay, this paper studies how the two interact. Appealing to previous works, they adapt the mirror flow framework for objectives with explicit regularization and use their new framework to demonstrate how explicit regularization can influence implicit bias. The key insight is that the implicit bias of the optimization algorithm can be adapted and controlled during the optimization procedure via dynamic explicit regularization. The paper demonstrates how the insights are manifest in experiments on sparse coding, matrix sensing, attention and LoRA fine-tuning illustrating potential applications.

## Update after rebuttal
After reading the reviews of others and seeing the rebuttals of the authors I have a better understanding of the work and its significance. I have appropriately moved my score up to a 3 (weak accept).

**Claims And Evidence:**

I am unsure whether the claims are well supported. In particular, the claims about 3 distinct effects of explicit regularization mentioned in the introduction. Moreover, the statement of the claims are unclear and not precise.

**Essential References Not Discussed:**

I am not familiar with this line of research to suggest any essential references not discussed.

**Experimental Designs Or Analyses:**

The experiments are sound although their implications are not clear.

**Methods And Evaluation Criteria:**

The paper is mainly a theoretical work but the evaluation criteria used in the experiments are appropriate. Although the applications are unclear.

**Other Comments Or Suggestions:**

### Minor
- Line 193, $R_a$ or $R_{a_t}$?

**Other Strengths And Weaknesses:**

### Strengths
The topic being studied is very interesting, novel, relevant and worthy of study. Extending the previous mirror flow framework to account for explicit regularization could also be of independent interest.

### Weaknesses
The biggest issue with the paper is presentation. I am not familiar with past work on mirror flow and this paper requires a lot of background knowledge about previous works connecting implicit bias of gradient descent to mirror flow. This makes it very difficult to understand the results and interpret their significance. The readability of the paper is also effected as many of the core details are left to the appendix and never formally defined. Moreover, the three different biases discussed in the introduction are also stated quite informally (except for the first) making it hard to really understand what they mean.

**Questions For Authors:**

1. By doing this dynamic explicit regularization, the objective function is changing for each $t$. Therefore, when the algorithm terminates, for which objective is $x_{*}$ a solution?

**Relation To Broader Scientific Literature:**

I am familiar with the idea of implicit bias and as far as I know it is typically studied independent of any explicit regularizers. Given that neural networks are often trained with explicit regularization, such as weight decay, I believe this paper brings insight to the important question of how explicit and implicit regularization interact and would be useful to the broader machine learning community.

**Theoretical Claims:**

The paper relies heavily on the setup and works of previous paper connecting the implicit bias of gradient descent to mirror flow. Largely building off of [1]. Without this background it is difficult to understand the setup or the motivation behind it. Moreover, too much of the technical details are left to the Appendix making it hard to read the paper. In light of this while I believe the theoretical claims are correct I am not familiar enough with this line of literature to verify their correctness.


[1] https://arxiv.org/abs/2207.04036

---

> ### Author Rebuttal · Authors · 2025-03-31
>
> We would like to express our gratitude for the reviewer’s time and effort in providing valuable comments on our manuscript and appreciate the acknowledgement of the work's relevance and novelty. We would be happy to discuss any open questions or concerns, if there remain any.
>
> **Background knowledge**
>
> We appreciate the reviewer’s observation regarding the significant background knowledge needed to fully understand the details. Given the initial page limit, we focused on presenting the main storyline.
>
> In the current manuscript, we elaborate on the implications with concrete examples in Section 4.
> Moreover, the main theorems are accompanied by detailed descriptions of the key proof steps, along with characterizations of the regularization and a geometric interpretation. These findings and concepts have been formally defined, highlighting important details of the analysis.
> As we would be allowed to use an additional page for the main manuscript upon acceptance, we would be happy to make adjustments. To this end, we will include more definitions from the appendix into the main manuscript like:
> 1. The definitions of a regular and commuting parameterization in Section 3 (Introduction part).
> 1. The definition of the Legendre function in Section 3 (Introduction part).
> 1. The definition of the Bregman function in Section 3.2.
>
> With these additions, we believe the main storyline would be accessible to a broader audience, with the key idea being that explicit regularization changes the implicit bias.
>
>
> **The 3 effects**
>
> We believe that the introduction provides a precise definition of the three effects without requiring the explicit introduction of the parameterized Legendre function. In Section 4, we present concrete theoretical examples of all three effects. However, we are happy to further clarify these effects by including the definition of the parameterized Legendre function at the beginning of Section 4:
> 1. Type of bias: The shape of $R_a$ changes with $a$.
> 1. Positional bias: The global minima of $R_a$ changes with $a$.
> 1. Range shrinking: The range of $\nabla R_a$ can shrink due to a specific choice of $a$.
>
> **Experimental implications**
>
> The experiments are there to verify the theory and to show that the principles hold in more practical settings. A practical implication is the potential utility of dynamic regularization schedules.
> For instance, in the case of quadratic reparameterizations, to get the best generalization accuracy for a desired sparsity ratio, it is beneficial to regularize more during the first half of training and then turn off weight decay in the second half. Moreover, our work provides a stepping stone for further research to derive better regularization schedules that take the implicit bias into account, as mentioned in the discussion.
>
> **Miscellaneous**
>
> We agree that it is more clear to use $R_{a_t}$ than $R_a$. The intent was to make a more direct reference to the previous definition of the parameterized Legendre function.
> In addition, $x_*$ is a solution of the objective $f$ as the regularization is turned off eventually. In other words, after turning off the regularization the iterates move to a solution of the objective $f$.
> We revise our manuscript accordingly.

---

### Official Review · Reviewer_Eyxr · 2025-03-16

**Overall Recommendation:** 3

**Summary:**

The paper investigates how external regularization influences the implicit bias of gradient flow. By leveraging the equivalence between parameterized gradient flow and mirror flow, the study provides a detailed analysis of how external regularization alters this mirror flow. Within the framework proposed by Li et al. (2023), when the parameterizing function $g$ and the regularization function $h$ are commuting re-parameterizations, the gradient flow on the regularized loss is shown to be equivalent to a mirror flow with a time-varying Bregman function. This Bregman function depends on regularization through the parameter $a$, which is defined as the integral of the regularization over time. The equivalence is used to study the convergence of regularized gradient flow and across different parameterizations or

**Claims And Evidence:**

a) The paper claims of different effects of regularization which it names as positional bias, type of bias and shrinking range which they claim as the main contribution of the paper. However, it feels that the the three effects are essentially the same, the range shrinking of $a$ changes the bias of $L_2$ to $L_1$ in quadratic parameterization.

**Essential References Not Discussed:**

To the best of my knowledge, the essential references have been sufficiently discussed.

**Experimental Designs Or Analyses:**

a) In the experiment corresponding to Figure 2, it is unclear how the regularization coefficient for the $ell_1$ is chosen, for appropriate choice I expect it to match the loss with regularization before turning it off. Also it is unclear comparison when both the regularization are turned off at the same iteration.

b) It is hard to interpret the experiment corresponding to Figure 3, if the higher nuclear norm to Frobenious norm ratio is desired, should the ideal method be to just choose a lower weight decay constant ?

**Methods And Evaluation Criteria:**

N/A as the paper is majorly theoretical in nature.

**Other Comments Or Suggestions:**

N/A

**Other Strengths And Weaknesses:**

Strengths :
- Theorem 3.2 is a nice addition to the literature of implicit bias and mirror flow showing how the weight decay modifies this revealing that the integral of the coeffecient across time is a key determining factor.

Weakness :
-  The framework is applied to different problems like matrix factorization and attention framework, however, the analysis holds when the matrices commuting. Hence, under the appropariate choice of eigenbasis, it is equivalent to the quadratic parameterization, it is recommended that the authors explicitly mention this.

**Questions For Authors:**

a) Theorem 3.6 uncovers an intriguing property: the integral of the regularization parameter is the sole determining factor. It would be nice to make this a bit concrete with experiments across some architectures, for example with various schedule which keep the integral to a constant value.

**Relation To Broader Scientific Literature:**

The equivalence between mirror flow and reparameterized gradient flow is extensively used to argue about the implicit bias of gradient methods with various hyperparameters  Woodworth et al., 2020; Pesme et. al. 2021. The work of Li et.al. 2022 lays the general framework when such a equivalence is possible and the current work extends this framework for regularized loss highlighting the differences.

**Theoretical Claims:**

a) In Theorem 3.2, the authors need to clearly specify that $(g,h) : M \to R^{n+1}$ should be a commuting parameterizations. In the current form, it reads as $g$, $h$ are commuting parameterization among themselves and not necessarliy $[ \nabla g_i, \nabla h]$ = 0

---

> ### Author Rebuttal · Authors · 2025-03-31
>
> We would like to express our gratitude for the reviewer’s time and effort in providing valuable comments on our manuscript and appreciate the acknowledgement of the theoretical contributions. Below, we address the raised questions and concerns, but would be happy to extend the discussion on request.
>
> **How the 3 effects are distinct**
>
> We provide some examples that highlight how the 3 effects are distinct from each other and also can occur independently from each other:
> 1. The range shrinking means that the actual range of $\nabla R_a$ changes during training; this does not occur for quadratic reparameterizations, for example.
> 2. The positional bias is distinct. When we initialize a quadratic parameterization at zero, the positional bias stays zero. So it does not change, while the type of bias changes still from L2 to L1.
> 3. The type of bias can be different in all these scenarios and even changes dynamically in most of our examples.
>
> **Commuting condition and eigenbasis**
>
> We will restate the commuting condition such that it is clear that indeed $(g,h)$ is commuting in Theorem 3.2. Moreover, we will mention the connection between an appropriate eigenbasis choice and the commuting condition explicitly after the quadratic parameterization result.
>
> **Figure 2,  L1 setting**
>
> The training dynamics for the linear parameterization are different from that of the quadratic reparameterization. Therefore, it is hard or impossible to match the same performance in the experiment.
> The main takeaway is that the type of implicit bias of the linear parameterization has not changed. In general, what would occur when the L1 regularization is turned off is that the training dynamics go to the L2 interpolation solution. To alleviate any concern, we have run an additional experiment for the L1 setting with higher regularization ($0.2$) and turned it off at the same time, this also leads to similar performance, i.e. it goes to the L2 interpolator with a reconstruction error of $0.32$. Therefore, the amount of regularization does not change the type of implicit bias. We will include this additional experiment in Appendix C of the revised manuscript.
>
>
> **Figure 3, ratio**
>
> We do not make claims about the desirability of these results; this would depend on the specific application. A lower rank (smaller ratio) might be desirable in cases where efficiency or interpretability is a priority, for example. For the practitioner it is relevant to understand how regularization impacts the rank and thus the solution. This insight is the purpose of our theory.
> The main message of this Figure 3 is that we can obtain similar ratios by turning off weight decay. Appendix F furthermore demonstrates that this can achieve better generalization. Specifically, at similar ratios, the validation accuracy of a ViT on ImageNet can be improved by more than 1%.
> This illustrates the utility of controllable implicit bias in practice, but the main purpose of our experiments is to verify our theory. We still believe our theory could inspire better algorithm design in future, as mentioned in the discussion.
>
> **Different schedules for matrix sensing**
>
> Consider the family of schedules with constant regularization strength $\alpha_i$ up to specific time T_i such that $\int_0^{T_i} \alpha_i ds = T_i \alpha_i = C$ for $C >0$ a constant. We choose $\alpha_i = [0.02, 0.2, 2, 20]$.  In addition, we consider a linear and cosine decay schedule for the regularization with the same total strength (i.e. same integral), but the regularization is switched off after half of the training time to ensure convergence. To compare with the effect of turning off (t-o) the regularization, we include the constant schedule with regularization strength $\alpha=0.01$. Note that this schedule is also reported in Appendix C. We observe that all schedules with decay or turn-off (t-o) converge to a solution with the same nuclear norm of the ground truth, confirming Theorem 3.6, while the constant schedule does not reach the ground truth. We would be happy to include this additional experiment in Appendix C.
>
> | Schedule                     | Nuclear norm | Train loss | Rec error | Time to 1e-7 train loss |
> |------------------------------|-------------|------------|-----------|--------------------------|
> | Constant 0.01, no t-o   | 0.93        | 7.2e-4     | 3.9e-2    | -                        |
> | Linear decay                 | 1.00        | 1.8e-8     | 2.3e-4    | 661                      |
> | Cosine decay                 | 1.00        | 1.7e-8     | 2.1e-4    | 624                      |
> | Constant 0.02, t-o      | 1.00        | 1.1e-8     | 1.7e-4    | 716                      |
> | Constant 0.2, t-o       | 1.00        | 2.7e-10    | 2.7e-5    | 209                      |
> | Constant 2, t-o         | 1.00        | 2.1e-10    | 2.4e-5    | 209                      |
> | Constant 20, t-o        | 1.00        | 7.9e-13    | 1.4e-6    | 239                      |

---

### Decision · Program_Chairs · 2025-05-01

**Decision:**

Accept (poster)

**Comment:**

This paper presents a novel theoretical analysis of how external regularization affects the implicit bias of gradient flow, by extending the mirror flow framework to account for time-varying Bregman functions induced by regularization. The reviewers appreciated the originality and relevance of the topic, as well as the insightful contribution of Theorem 3.2, which reveals the role of WD through integrated regularization coefficient. While the presentation could be improved, particularly for readers unfamiliar with prior mirror flow work, the technical depth and potential impact on our understanding of implicit bias are significant. I recommend acceptance.